# Loss of Gap Junction Delta-2 (*GJD2*) gene orthologs leads to refractive error in zebrafish

Wim H. Quint [1,2✉], Kirke C. D. Tadema[1,2], Erik de Vrieze [3], Rachel M. Lukowicz[4], Sanne Broekman[3],
Beerend H. J. Winkelman[1,5], Melanie Hoevenaars[1,2], H. Martijn de Gruiter [6], Erwin van Wijk[3],
Frank Schaeffel[7,8], Magda Meester-Smoor[1,9], Adam C. Miller [4], Rob Willemsen[2],
Caroline C. W. Klaver [1,8,9,10] & Adriana I. Iglesias [1,2✉]

Myopia is the most common developmental disorder of juvenile eyes, and it has become an increasing cause of severe visual impairment. The *GJD2* locus has been consistently associated with myopia in multiple independent genome-wide association studies. However, despite the strong genetic evidence, little is known about the functional role of *GJD2* in refractive error development. Here, we find that depletion of *gjd2a* (Cx35.5) or *gjd2b* (Cx35.1) orthologs in zebrafish, cause changes in the biometry and refractive status of the eye. Our immunohistological and scRNA sequencing studies show that Cx35.5 (*gjd2a*) is a retinal connexin and its depletion leads to hyperopia and electrophysiological changes in the retina. These findings support a role for Cx35.5 (*gjd2a*) in the regulation of ocular biometry. Cx35.1 (*gjd2b*) has previously been identified in the retina, however, we found an additional lenticular role. Lack of Cx35.1 (*gjd2b*) led to a nuclear cataract that triggered axial elongation. Our results provide functional evidence of a link between *gjd2* and refractive error.

[1] Department of Ophthalmology, Erasmus Medical Center, Rotterdam, The Netherlands. [2] Department of Clinical Genetics, Erasmus Medical Center, Rotterdam, The Netherlands. [3] Department of Otorhinolaryngology, Donders Institute for Brain, Cognition and Behavior, Radboud University Medical Center, Nijmegen, Netherlands. [4] Institute of Neuroscience, University of Oregon, Eugene, United States. [5] Department of Cerebellar Coordination and Cognition, Netherlands Institute for Neuroscience, Amsterdam, The Netherlands. [6] Optical Imaging Centre, Erasmus Medical Center, Rotterdam, The Netherlands. [7] Institute for Ophthalmic Research, University of Tübingen, Tübingen, Germany. [8] Institute of Molecular and Clinical Ophthalmology Basel, Basel, Switzerland. [9] Department of Epidemiology, Erasmus Medical Center, Rotterdam, The Netherlands. [10] Department of Ophthalmology, Radboud University Medical Center, Nijmegen, The Netherlands. ✉email: w.quint@erasmusmc.nl; a.iglesiasgonzalez@erasmusmc.nl

Refractive errors (REs) arise when the focal length of the optics of the eye are not matched to the length of the eye, causing a defocused image on the retina. Myopia has become the most common type of RE after a significant increase in prevalence over the last decades[1]. Given the current trends, 50% of the world population is expected to have myopia by the year 2050, and 10% will have high myopia with REs of −6 diopter (D) or higher[1]. The rising global prevalence drives a serious burden as myopia is linked to an increased risk of common and often sight-threatening eye diseases such as myopic macular degeneration, glaucoma, retinal detachment, and cataract[2,3].

Myopia is a complex trait influenced by an interplay of both genetic and environmental factors. Genome-wide association studies (GWAS) have been successfully used to understand the genetic background of RE. In 2010, one of the first GWAS for RE in European populations identified a locus near the gap junction protein delta-2 (GJD2) gene, harboring regulatory elements that could potentially influence the transcription of GJD2[4]. After this initial discovery, the association of this locus to myopia has been replicated by multiple independent studies in numerous ethnicities[5–15]. Despite the strong genetic evidence of an RE locus in the vicinity of GJD2, little is known about the functional role of GJD2 in RE.

GJD2 encodes the 36-kDa protein connexin 36 (Cx36), a member of the connexin (Cx) protein family, and a key element of neuronal gap junctions. Of note, throughout the manuscript, we will use the official gene name (e.g., GJD2) when referring to the gene and protein name (e.g., Cx36) when referring to the protein. Within the nervous system, gap junctions function as electrical synapses regulating the bidirectional flow of ions and small metabolites between various neural cell types[16,17]. Although Cx36 is known to play an essential role in electrical gap junction signaling[16,18,19] in the majority of retinal cell types[16,20–31], the molecular mechanism underlying the potential function of Cx36 gap junctions in regulating emmetropization remains unknown.

In this study, we used zebrafish (Danio rerio) to gain insight into potential mechanisms for GJD2 in RE. We examined the ocular consequences of the loss of function of two GJD2 (Cx36) zebrafish homologs: gjd2a (Cx35.5) and gjd2b (Cx35.1). Both Cx35.5 (gjd2a) and Cx35.1 (gjd2b) are known to assemble at the electrical synapse across the zebrafish central nervous system (CNS)[32–34]. The generally held view is that Cx35.1 (gjd2b) acts as the main retinal connexin, while Cx35.5 (gjd2a) functions mainly in the CNS[32,35]. Our results suggest that both gjd2a (Cx35.5) and gjd2b (Cx35.1) are expressed in the retina, and that loss of function leads to alterations in ocular biometry and development of RE.

## Results

### Loss of Cx35.5 (gjd2a), but not Cx35.1 (gjd2b), led to reduced ocular dimensions.

We examined zebrafish from the juvenile stage (1.5 and 2 months post-fertilization (mpf)) into adulthood (3 mpf) and studied two independent lines that were deficient for Cx35.5 (gjd2a) or Cx35.1 (gjd2b)[32]. To prevent biases induced by potential differences in the mean body sizes of gjd2a (Cx35.5) and gjd2b (Cx35.1) mutants, we compared these fish to wild type (WT) controls that matched the mean body length of the mutant group within a 10% range (see details in "Methods", Supplementary Fig. S1). We measured the ocular biometry with spectral-domain optical coherence tomography (SD-OCT) and defined axial length as the distance from the apical part of the corneal epithelium to the anterior part of the retinal pigmented epithelium (RPE). Figure 1a shows a representative SD-OCT image of a zebrafish eye.

In the gjd2a (Cx35.5) mutants, we observed a significant reduction in axial length (Fig. 1b, Supplementary Data S1). This reduction was mainly due to a decrease in lens diameter and vitreous chamber depth (Fig. 1d, e). By contrast, no alterations in axial length were observed in the gjd2b (Cx35.1) mutants (Fig. 1c, Supplementary Data S2). Both gjd2a (Cx35.5) and gjd2b (Cx35.1) mutants showed alterations in corneal thickness, anterior chamber depth, and RPE thickness (Supplementary Data S1 and 2, Fig. 1f), however, given the relatively thin cornea (~20–26 μm), RPE thickness (~24–36 μm), and almost non-existent anterior chamber depth (~4–6 μm) (Fig. 1a), measurements were potentially prone to a higher level of inaccuracy.

### Loss of Cx35.5 (gjd2a) or Cx35.1 (gjd2b) changed RE.

To determine whether the observed alterations in axial length led to RE, we assessed the refractive status of all lines using eccentric photorefraction. Eccentric photorefraction has served as a sensitive method for refractive measurements in many species[36–38], however, it has not been described for zebrafish yet. Here, we adjusted the eccentric photorefractor setup for use in zebrafish eyes (see "Methods"). Figure 2 shows the differences in RE between mutant and WT fish. The baseline WT RE appeared to be hyperopic (Fig. 2d, e), an effect induced by the small eye retinoscopic artifact as described previously[36,39,40]. We visualized the relative differences in RE between mutant and control fish (Fig. 2f). Consistent with their reduced axial length, the gjd2a (Cx35.5) mutants showed a progressive positive (hyperopic) RE (Fig. 2d, f). Given that the anteroposterior lens diameter in the gjd2a (Cx35.5) mutants was reduced and that thinner or flattened lenses may also lead to hyperopic defocus[41], we isolated the lenses of gjd2a (Cx35.5) fish (n = 10) but did not observe any macroscopic abnormalities in lens shape and curvature compared with lenses of WT fish.

In contrast to gjd2a (Cx35.5), in the gjd2b (Cx35.1) mutant eyes, we found a progressive negative (myopic) RE (Fig. 2e, f) even though we did not observe alterations in axial length (Fig. 1c). Interestingly, at 9mpf, we noticed that the gjd2b (Cx35.1) mutants showed a relatively large variation in RE measurements (relative RE ranging between −2D and −20D, Fig. 2f). We found that a small proportion (n = 3/20) of gjd2b (Cx35.1) mutant eyes showed no visible intensity gradient, a phenomenon we observed in fish with severely enlarged axial lengths (e.g., in the Bugeye (lrp2−/−) mutant[42,43]). We measured a group of lrp2 mutants with moderate axial myopia and found a typical myopic intensity gradient, whereas the gjd2b (Cx35.1) fish displayed a seemingly bifocal signature (Fig. 2c).

### Loss of Cx35.1 (gjd2b) induced nuclear cataract underlying lenticular myopia.

To study the origin of the non-axial form of myopia and bifocal intensity gradient found in the gjd2b (Cx35.1) mutants, we examined the lenticular appearance of the 1.5, 2, and 3 mpf fish on SD-OCT. We were able to observe the nuclear fiber structures in the gjd2b (Cx35.1) mutant lenses (Fig. 3b), which were not visible in WT (Fig. 3a) and gjd2a (Cx35.5) mutant fish (Supplementary Fig. S2). The location of the opacities showed resemblance with OCT images of human nuclear cataract patients[44–48]. We quantified the proportion of eyes with nuclear opacities and found a 25% increase of cataractous lenses at 1.5–2 mpf and a 60% increase in 3 mpf gjd2b (Cx35.1) mutants, relative to WT fish (Fig. 3c). To explore the potential progression of cataractous lenses into adulthood, we assessed an additional group of 6 mpf fish using SD-OCT and differential interference contrast (DIC) microscopy (Fig. 3d–g). At this age, using SD-OCT, 92% of the lenses of the gjd2b (Cx35.1) mutants showed opaque ring-like fiber structures, whereas this was not visible in WT lenses (Fig. 3h). Ex vivo DIC microscopy validated the SD-OCT findings and showed clear visualization of the nuclear fiber

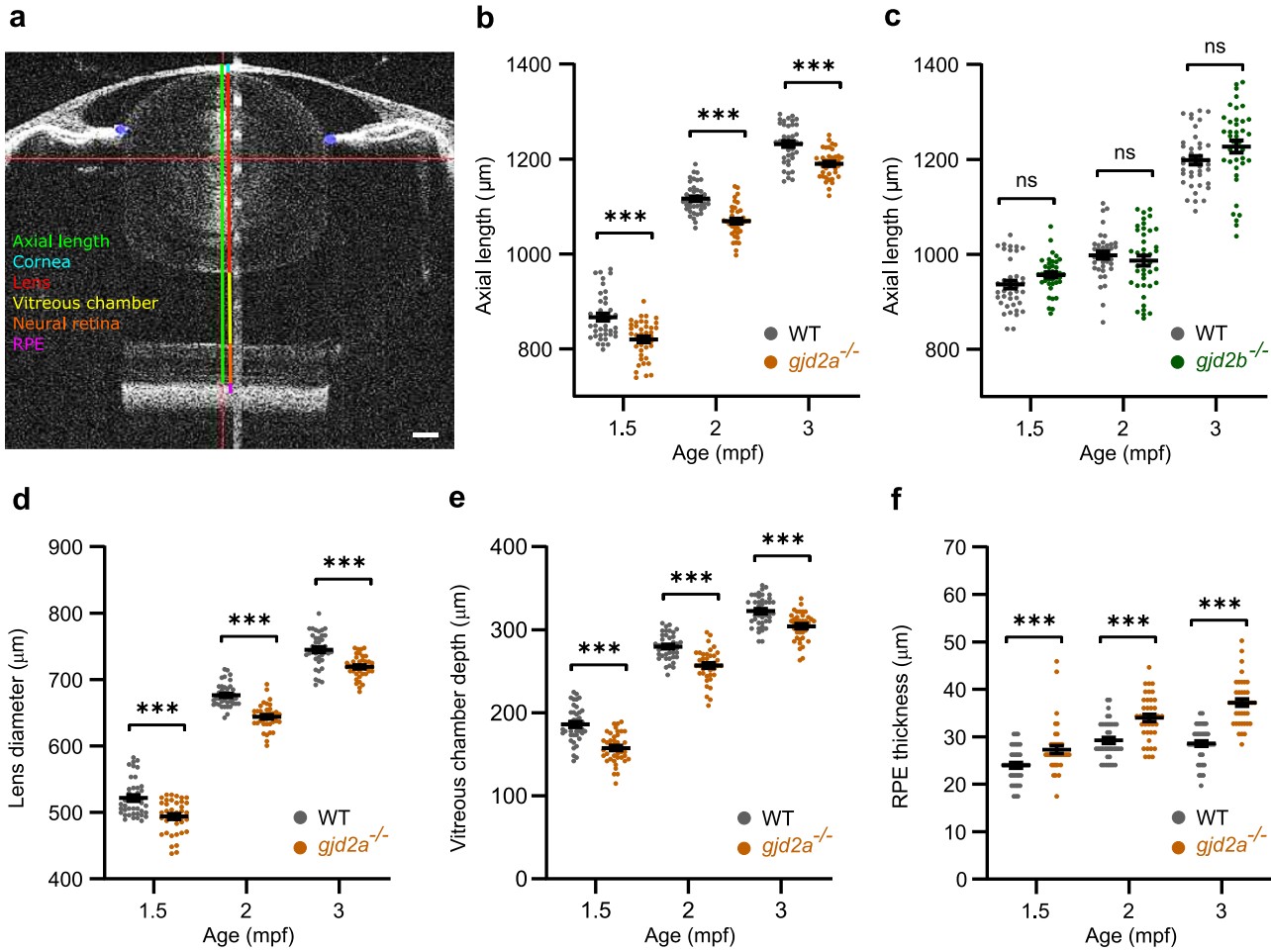

**Fig. 1 Loss of Cx35.5 (*gjd2a*) leads to reduced ocular dimensions.** SD-OCT recordings of size-matched juvenile-to-adult zebrafish indicating the temporal dynamics of the eye. All ocular metrics were corrected for the tissue-specific refractive index. **a** Single B-scan image of a 3 mpf zebrafish eye. The area is defined as the axial length that spans from the apical part of the corneal epithelium to the anterior border of the RPE (green line). The RPE is represented by the hyperreflective melanin-rich band (magenta), of which the anterior part comprises a sharp-cut border, used as a posterior landmark for the axial length. The gradient refractive index of the spherical zebrafish lens was used as a correction factor to acquire this image (see "Methods"), and the brightness was enhanced for better visualization of the transparent lens. Individual compartments: cornea (light blue), lens (red), vitreous chamber (yellow), neural retina (orange). **b**, **c** Axial length of *gjd2a* (Cx35.5) (**b**) and *gjd2b* (Cx35.1) (**c**) mutant eyes of juvenile (1.5–2 mpf) and adult zebrafish (3 mpf). **b** The *gjd2a* (Cx35.5) eyes were significant reduced in axial length compared with WT at 1.5 mpf (effect size = −47 μm, p < 0.001), 2 mpf (Effect size = −48 μm, p < 0.001), and 3 mpf (effect size = −43 μm, p < 0.001). **c** The *gjd2b* (Cx35.1) mutant eyes showed no significant alteration relative to WT. **d**, **e**, **f** Dimensions of significantly altered ocular compartments in *gjd2a* (Cx35.5) mutant eyes. See Supplementary Data S1 and S2 for a full statistical report and dimensions of individual compartments. Sample size: n = 40 eyes for each genotype and time point. Error bars: SEM. Significance: ns = not significant, *p < 0.05, **p < 0.01, ***p < 0.001. Scale bar: 100 μm. Mpf months post-fertilization, SD-OCT spectral-domain optical coherence tomography, RPE retina pigmented epithelium.

structure in all examined lenses of *gjd2b* (Cx35.1) mutants (Fig. 3e, g, h), whereas this was only modestly visible in 8% of the WT lenses (Fig. 3d, f, h). Figure 3i shows that the proportion of opaque pixels in SD-OCT images of 6mpf lenses was increased by 25% in *gjd2b* (Cx35.1) mutants.

Unexpectedly, we observed that the highly cataractous 6 mpf *gjd2b* (Cx35.1) mutants also had a significant increase (7%) in total axial length (Fig. 4a, Supplementary Data S3), RPE thickness (27%), lens diameter (6%), and vitreous chamber depth (4%) (Supplementary Data S3). To investigate whether lens blurring and subsequent attenuation of retinal image quality could be a cause of the observed eye growth, we created an ex vivo setup to assess the basic optical properties of the lens. We detected a significant reduction in the translucent properties of the *gjd2b* (Cx35.1) mutant lenses for a variety of wavelengths (365 nm, 940 nm, and 380–760 nm) (Fig. 4b). To further evaluate this, we visualized the light propagation and potential image distortion of

isolated 6mpf lenses ex vivo (Fig. 4c). We observed light scattering and multifocality by two moderately cataractous lenses (Fig. 4e, f) and profound light diffusion by a severely cataractous lens (Fig. 4g) of the *gjd2b* (Cx35.1) mutant. The light path of the WT control lenses (n = 3) showed a finer distinction of individual light rays and more homogenous dispersion (Fig. 4d). This observation confirmed that the cataract indeed had the potential of reducing image contrast.

**B-wave potential and spatial acuity in Cx35.5 (*gjd2a*) and Cx35.1 (*gjd2b*) mutants.** To explore the functional consequences of the loss of Cx35.5 (*gjd2a*) and Cx35.1 (*gjd2b*) on retinal light processing, we measured the electrical potential of the retina using electroretinograms (ERG) focusing on the B-wave. In total, 2.5 mpf *gjd2a* (Cx35.5) eyes showed a significant decrease in B-wave amplitude compared to WT control (Fig. 5a, b). To examine

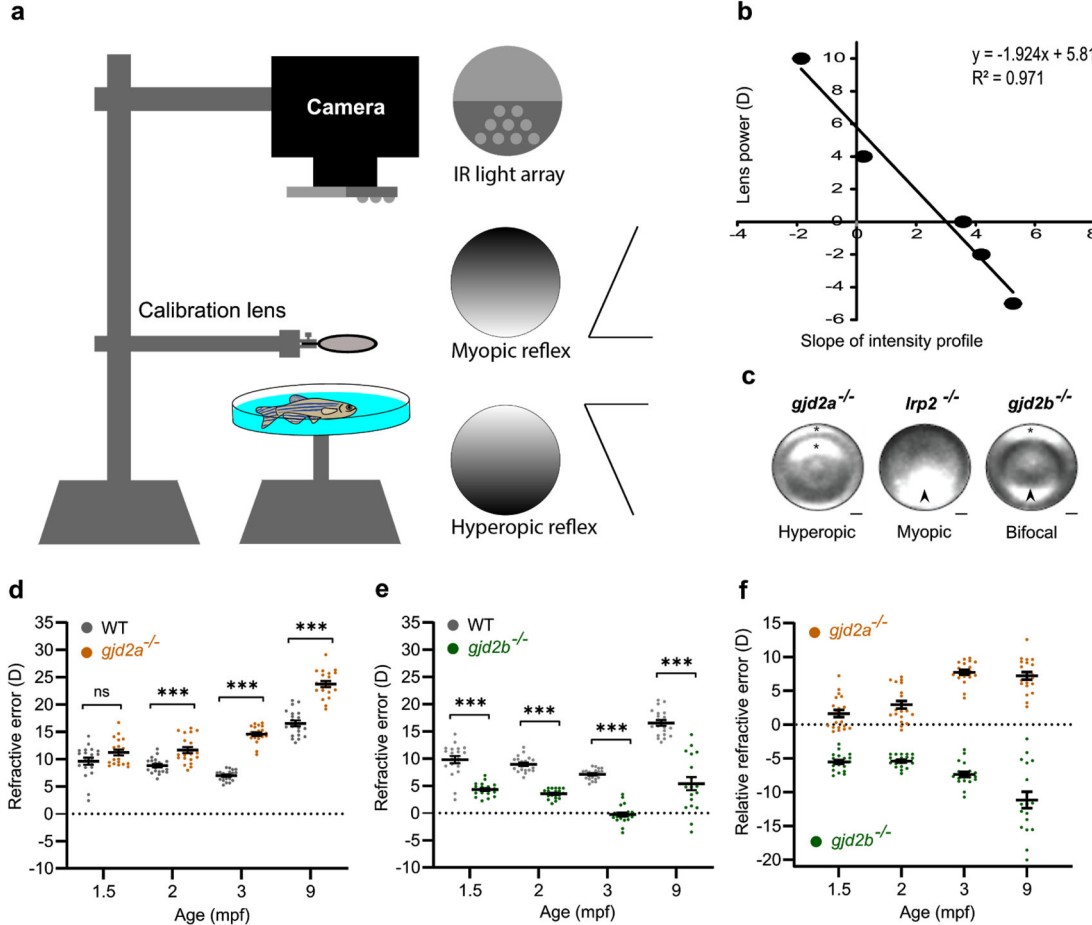

**Fig. 2 Opposite refractive error in *gjd2a* (Cx35.5) and *gjd2b* (Cx35.1) mutants. a** Schematic illustration of the eccentric photorefractor setup. **b** Calibration by −6 diopter (D), −2D, 0D, +4D, and +10D lenses led to a conversion factor of 1.924 ($R^2 = 0.971$). **c** Typical intensity profile of a hyperopic (asterisk) 3mpf *gjd2a* (Cx35.5) mutant and myopic (arrowhead) *lrp2* mutant. The *gjd2b* (Cx35.1) mutant shows both myopic (arrowhead) and hyperopic (asterisk) contralateral features. **d**, **e** RE in the *gjd2a* (Cx35.5) (**d**) and *gjd2b* (Cx35.1) (**e**) mutants at 1.5 mpf, 2 mpf, 3 mpf, and 9 mpf. **d** Loss of Cx35.5 (*gjd2a*) results in a significant ($p < 0.001$) and progressive hyperopic shift in refractive status. **e** Loss of Cx35.1 (*gjd2b*) is linked to a significant ($p < 0.001$) and progressive myopic shift. **f** Mutant refractive status normalized against the baseline refraction of WT controls, indicated by the relative RE. Sample size: $n = 20$ eyes for each genotype and age. Error bars: SEM. Significance: ns = not significant, *$p < 0.05$, **$p < 0.01$, ***$p < 0.001$. Scale bars: 50 µm (**c**). RE refractive error.

whether body length influenced the B-wave amplitude, we corrected the maximum B-wave amplitude for body length and still found a significant decrease in amplitude (Supplementary Fig. S3). The B-wave potential of the 2.5 mpf *gjd2b* (Cx35.1) mutants was not different from WT fish (Fig. 5a, b).

In 3 mpf fish, we examined the spatial acuity by the optokinetic response (Supplementary Fig. S4c). The proportion of positive responders reacting to the stimulus with three subsequent correct responses (Fig. 5c) was equal in all fish lines, as was the optokinetic gain for spatial frequencies ranging from 0.15 to 0.25 cpd (Supplementary Fig. S4a, b). In contrast, a lower positive response tendency was observed for stimuli above 0.25cpd in both the *gjd2a* (Cx35.5) and *gjd2b* (Cx35.1) mutants (Fig. 5c). The *gjd2b* (Cx35.1) mutants also showed a significant reduction in nasally directed eye tracking movements per 10-s interval (Fig. 5d).

**Identification of Cx35.5 (*gjd2a*) and Cx35.1 (*gjd2b*) in ocular tissue**. To investigate the expression of Cx35.5 (*gjd2a*) protein in retinal tissue, we performed immunostaining using a specific anti-Cx35.5 (*gjd2a*) monoclonal antibody[32] in 2.5 mpf fish. In WT fish, we observed a punctate signal throughout the neural retina

(Fig. 6e–h). Modest Cx35.5 (*gjd2a*) plaques were visible in the somata of retinal ganglion cells and more abundantly in the inner plexiform layer (IPL) and outer plexiform layer (OPL). In addition, in the outer retina, a modest distribution of small gap junction plaques was found between adjacent photoreceptors. The signal was absent in the *gjd2a* (Cx35.5) mutants, confirming the specificity of the antibody (Fig. 6m–p).

We also used a commercially available antibody MAB3045 designed against recombinant perch Cx35 protein[49,50], which was reported to bind to connexin 35/36 and to show cross-reactivity with Cx35.5 (*gjd2a*) in the zebrafish CNS[32]. As expected, we found a typical staining pattern of gap junction plaques in the WT retina (Fig. 6a–d) and further observed remnant immunostaining in both *gjd2a* (Cx35.5) (Fig. 6i–l) and *gjd2b* (Cx35.1) (Fig. 6q–t) mutants. To rule out that the MAB3045 antibody cross-reacted with other related retinal connexins, i.e., Cx34.1 (*gjd1a*) and Cx34.7 (*gjd1b*)[29,32,34,49,51], we examined the staining profile in a mutant for both *gjd2* paralogues (*gjd2a*; *gjd2b* double mutant). A complete loss of immunostaining in the *gjd2a*; *gjd2b* double mutant (Supplementary Fig. S5c, d, Supplementary Table S1) confirmed the specificity of the MAB3045 antibody for Cx35.5 (*gjd2a*) and Cx35.1 (*gjd2b*). Hence, we refer to this antibody as the "anti-pan-Cx35".

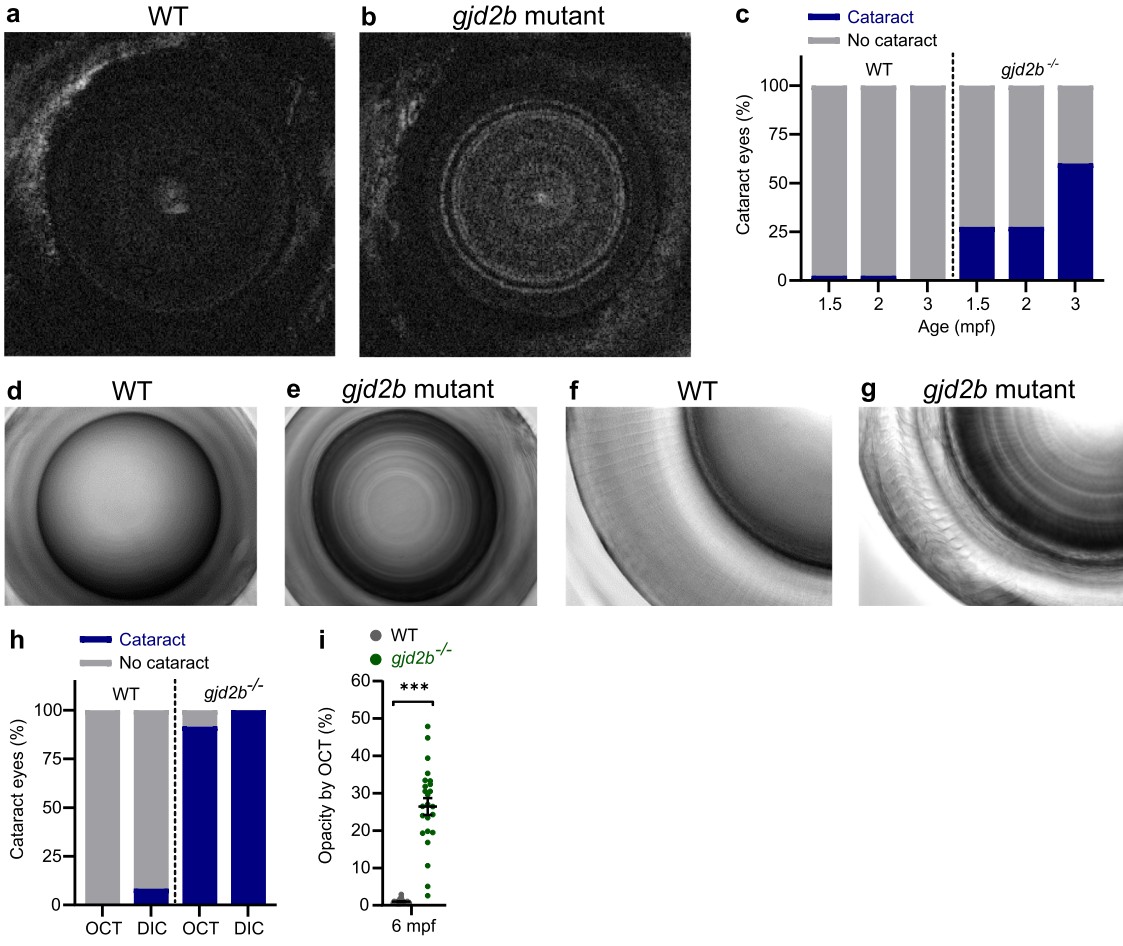

**Fig. 3 Nuclear cataract in *gjd2b* (Cx35.1) mutants. a**, **b** Coronal SD-OCT sections of typical 3 mpf lenses of WT control (**a**) and *gjd2b* (Cx35.1) mutant (**b**) fish. **c** Proportion of cataractous lenses in 1.5 mpf, 2 mpf, and 3 mpf SD-OCT data ($n = 40$ eyes) indicating an increasing prevalence in *gjd2b* (Cx35.1) mutants. **d**–**g** DIC microscopy of 6 mpf fish lenses allows visualization of opaque nuclear lens fibers in lenses of *gjd2b* (Cx35.1) mutants (**e**) and (**g**) and transparent WT control lenses (**d**) and (**f**). **d**, **e** DIC 20× magnification (**f**) and (**g**) 40× magnification. **h** Proportion of cataractous lenses by SD-OCT and DIC microscopy at 6 mpf ($n = 24$ eyes). **i** Ratio of opaque pixels: total pixels indicating the opacity of 6 mpf coronal SD-OCT sections ($n = 24$ eyes). Error bars: SEM. Significance: ns = not significant, *$p < 0.05$, **$p < 0.01$, ***$p < 0.001$. DIC microscopy: differential interference contrast microscopy.

The combination of immunostainings for anti-pan-Cx35 and anti-*gjd2a*/Cx35.5 in both mutants (Supplementary Table S1) showed the layer-specific topographic distribution of the Cx35.5 (*gjd2a*) and Cx35.1 (*gjd2b*) gap junctions. In the IPL, the typical punctate staining pattern marked gap junction plaques containing both homologs. In the outer plexiform layer, we found a relatively dense population of Cx35.5 (*gjd2a*) and Cx35.1 (*gjd2b*) plaques. Furthermore, we observed modest Cx35.5 (*gjd2a*) and Cx35.1 (*gjd2b*) gap junction plaques in the photoreceptor layer between the outer segments as well as across the outer nuclear layer (Fig. 6c, g, k, s).

Given the cataractous phenotype of the *gjd2b* (Cx35.1) mutant fish, we investigated the distribution of Cx35.1 (*gjd2b*) in the lens. Immunostaining of adult WT lenses with anti-pan-Cx35 resulted in modest lateral immunostaining around the outer cortical region; this was lacking in the *gjd2b* (Cx35.1) mutant. The immunostaining for anti-pan-Cx35 in lenses of the *gjd2a* (Cx35.5) mutant was similar to the WT lenses (Supplementary Fig. S6).

**Cell-specific expression in larval retinal tissue.** To further investigate the expression of retinal *gjd2a* (Cx35.5) and *gjd2b* (Cx35.1), we examined single-cell RNA sequencing (scRNAseq) from a whole-embryo dataset[52]. Relevant retinal cells were

captured from the larger dataset using the expression of established tissue and cell markers for retinal cell types (Supplementary Data S4), and the resultant dataset encompassed a total of 2218 cells spanning 2–5 days post-fertilization (dpf) fish (Fig. 7a, Supplementary Fig. S7), a critical window of eye development. At this larval stage, we detected a modest expression of both *gjd2a* (Cx35.5) and *gjd2b* (Cx35.1) throughout a variety of retinal cell types (Fig. 7b, c). Only a small fraction of cells were found to be positive for the paralogues (Fig. 7b, c), however, the detected expression overlapped across most putative cell types for both genes (Fig. 7b, c). At this early time point, transcripts for both genes were detected in a small number of rods and cones, bipolar cells, amacrine cells, and, to a lesser extent, Müller glia. These were consistent with the immunostaining observed in the OPL and IPL (Fig. 6). In addition, we found a cluster of putative retinal ganglion cells expressing *gjd2a* (Cx35.5), whereas we did not detect *gjd2b* (Cx35.1) expression in ganglion cells, while putative horizontal cells expressed *gjd2b* (Cx35.1) but not *gjd2a* (Cx35.5) (Fig. 7b, c). We note that the histological and scRNAseq datasets derive from different points in development, and may account for differences observed, particularly the low levels of expression detected for *gjd2a* (Cx35.5) in the scRNAseq data. In short, the histological and scRNAseq findings showed a corresponding expression pattern for both *gjd2a* (Cx35.5) and *gjd2b* (Cx35.1).

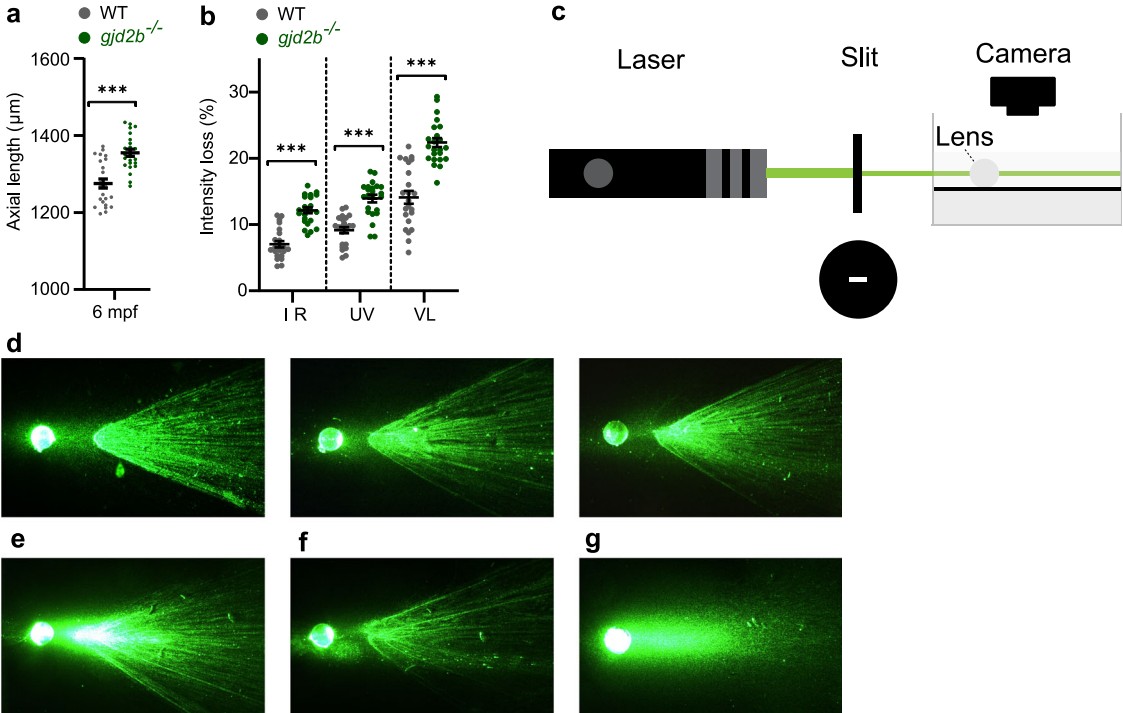

**Fig. 4 Late-onset axial growth in *gjd2b* (Cx35.1) mutants is induced by retinal image degradation. a** SD-OCT shows a significant increase in axial length in 6 mpf *gjd2b* (Cx35.1) mutants (*n* = 24 eyes). **b** Loss of lenticular projection intensity of isolated mutant lenses (*n* = 24 eyes), the following wavelengths were used: IR light (940 nm), UV light (365 nm), and broad-spectrum visible light (380–760 nm). **c** Schematic illustration of the ex vivo lenticular light propagation setup. **d** Examples of light propagation in three isolated 6 mpf WT control lenses. **e–g** Examples of light scattering and multifocality by two moderately cataractous lenses (**e**) and (**f**) and light diffusion by a severely cataractous lens (**g**) of the 6 mpf *gjd2b* (Cx35.1) mutant. Error bars: SEM. Significance: ns = not significant, \**p* < 0.05, \*\**p* < 0.01, \*\*\**p* < 0.001. IR infrared, UV ultraviolet.

## Discussion

In this study, we aimed to increase insight into the potential role of *GJD2* in human RE. We performed an in-depth ocular characterization of two *GJD2* (Cx36) zebrafish orthologs: *gjd2a* (Cx35.5) and *gjd2b* (Cx35.1). We compared the biometry, optics, and electrophysiology of mutant zebrafish eyes with WT eyes, and characterized protein localization by histological and gene expression analyses. Our results suggest that depletion of the orthologs leads to various ocular phenotypes, including biometrical, optical, structural, and electrophysiological changes.

In the *gjd2a* (Cx35.5) mutant, we found a hyperopic phenotype mostly determined by a reduction in axial length. The main ocular components that underlined the axial reduction were decreased vitreous chamber depth and lens diameter. Furthermore, our ERG studies showed a reduction in the B-wave amplitude of these mutant fish. This is the first time that biometrical changes and RE have been characterized in a connexin depleted animal model; however, the diminished B-wave potentials have been reported earlier in *Gjd2* (Cx36) null mice and have been attributed to defects in the ON-bipolar signaling[53–57]. It remains unknown why depletion of *gjd2a* (Cx35.5) was sufficient to provoke an electrophysiological effect on ERG while not significantly altering the spatial acuity measured by the optokinetic response, as ON-bipolar cell signaling is also highly involved in spatial vision[58]. Besides, the *gjd2a* (Cx35.5) mutant fish showed a hyperopic phenotype, but no other eye abnormalities were observed. This finding is in line with what is expected from a gene involved in regulating axial eye growth. It is worth mentioning that in humans, *GJD* (Cx36) is associated with RE in the general population and not with syndromic forms of myopia. Therefore, alterations in axial length without other gross abnormalities resemble the human situation.

The precise molecular mechanism explaining the relation between retinal gap junction coupling and eye growth remains unknown. However, based on our results in the *gjd2a* (Cx35.5) mutant fish and the reported literature, we hypothesize that the uncoupling of retinal gap junctions inhibits ocular growth. Evidence supporting this notion includes the observation that pharmacological uncoupling of gap junctions using meclofenamic acid protects against form-deprivation myopia (FDM), an experimental form of myopia, in chicks[59,60]. Furthermore, it has been shown that high-intensity light has a protective effect against childhood[61–64] and experimental[60,65–67] myopia, whereas chicks and monkeys exposed to continuous high-intensity light developed severe hyperopia[67,68]. It is known that lighting conditions can modulate retinal gap junction coupling through various pathways and neuromodulators by modifying its phosphorylation state[28,59,60,69]. For example, it has been described that Cx35/Cx36 remains in a dephosphorylated (uncoupled) state under bright lighting conditions[56,70–73]. Therefore, we speculate that high-illuminant light inhibits ocular growth through the uncoupling of retinal gap junctions. Two important modulators are dopamine and adenosine. Light-directed Cx35/36 (de)phosphorylation has been described to be regulated by dopamine signaling (effect: dephosphorylation/uncoupling)[73–82] and adenosine signaling (effect: phosphorylation/coupling)[70,71,82]. Furthermore, studies showed that adenosine antagonists (uncoupling effect) appeared effective against childhood[83] and experimental[84–86] myopia, whereas stimulating dopamine signaling (uncoupling effect) in the eye appeared protective against experimental myopia[59,68,74–81,87,88]. In short, the previous findings suggest that the uncoupling of retinal gap junctions through either meclofenamic acid, high-intensity light, or adenosine inhibition/dopamine stimulation all have inhibitory

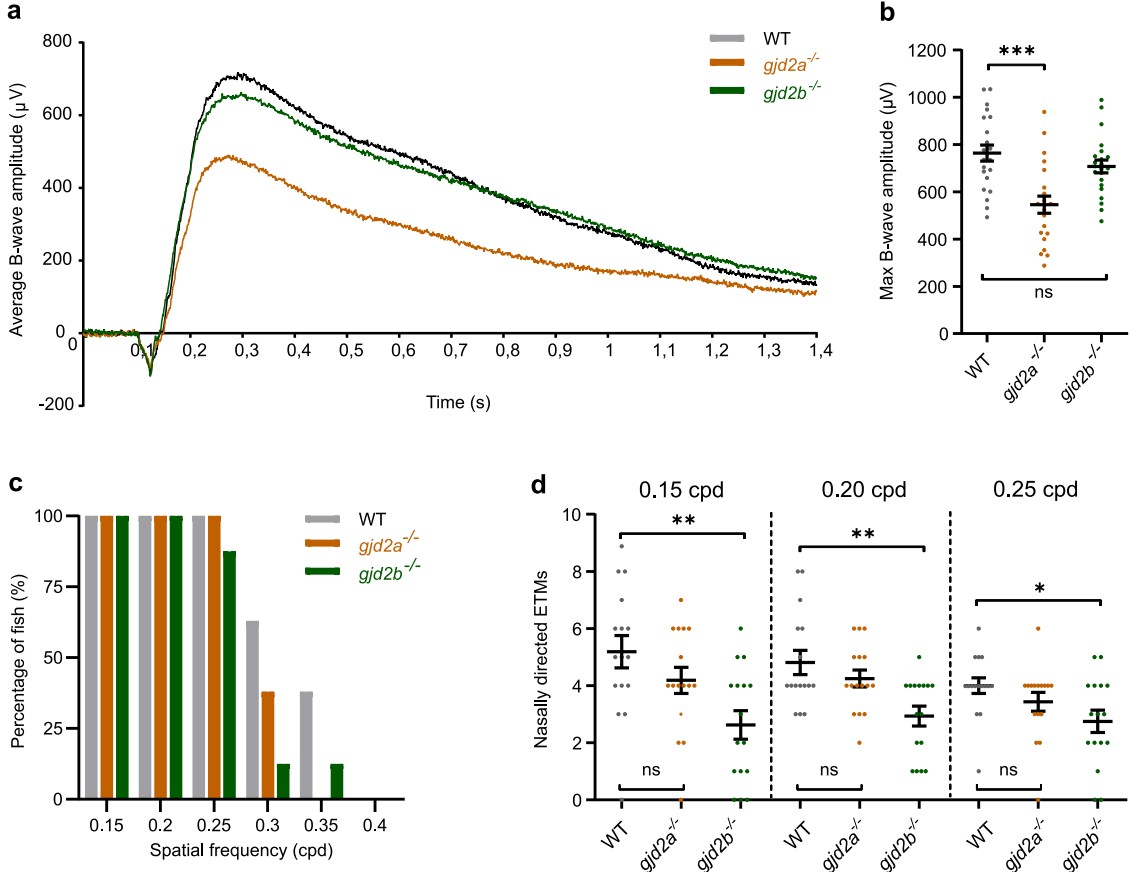

**Fig. 5 B-wave potential and spatial acuity in *gjd2a* (Cx35.5) and *gjd2b* (Cx35.1) mutants. a** Electroretinogram showing the average B-wave potential for eyes of *gjd2a* (Cx35.5) mutants, *gjd2b* (Cx35.1) mutants, and WT control fish at 2.5 mpf (*n* = 22 fish). **b** Maximum B-wave amplitude response. **c** Spatial visual acuity indicated by the proportion of fish completing a minimum of three subsequent optokinetic responses at spatial frequencies ranging from 0.15 to 0.40 cpd. **d** Nasally directed ETMs per 10-s interval for spatial frequencies ranging from 0.15 to 0.25 cpd. Error bars: SEM. Significance: ns = not significant, *p < 0.05, **p < 0.01, ***p < 0.001. ETM eye-tracking movements, cpd cycles per degree.

effects on axial growth (Fig. 8). Similarly, the axial reduction in the *gjd2a* (Cx35.5) mutants may have resulted from the permanent ablation of Cx35.5 (*gjd2a*) gap junctions in the retina, leading to a lower degree of cellular coupling in the retina.

A limited number of studies have explored the phosphorylation state of Cx36 in animal models of myopia. A recent study describes an FDM murine model in which eyelids were sutured for 40 days. After removing the lid suture, mice showing a negative RE were selected for immunohistochemistry studies in which the phosphorylation state of Cx36 was assessed. The authors report increased phosphorylation and coupling of Cx36 between AII amacrine cells in the studied eyes and attribute their finding to a compensation mechanism aiming to improve the signal-to-noise ratio caused by the defocused state during lid suture[89]. Other studies exploring the intricate relation between the phosphorylation state of Cx36 (*GJD2*) in the retina and RE are warranted.

In the *gjd2b* (Cx35.1) mutant, we observed a normal axial length; however, unexpectedly, a negative (myopic) RE by photorefraction. In our photorefraction studies, we discovered that the intensity gradient of the *gjd2b* (Cx35.1) mutants showed a bifocal signature when compared to the myopic *lrp2* mutants. By SD-OCT and DIC, we found that the RE in the *gjd2b* (Cx35.1) mutants had a lenticular origin due to a progressive, cataract-like, opacification of the nuclear layers. In the mammalian lens, gap junctions include connexin 43 (Cx43), 46 (Cx46), and 50 (Cx50)[90–93]. Consistent with our results in the *gjd2b* (Cx35.1)

mutant, the loss of mammalian Cx46 and Cx50 also leads to cataract development[90–93], and mutations in human Cx46 and Cx50 have been associated with autosomal dominant congenital cataract[94–97]. Although to date, Cx35.1 (*gjd2b*) has not been described as a lenticular connexin, four other main connexins, Cx43, Cx44.1, Cx48.5[98,99], and the larger Cx79.8[29], have previously been found in the lens. Whereas Cx48.5 zebrafish morphants showed modest nuclear opacities at 9.5 dpf[99], the *gjd2b* (Cx35.1) mutant showed a cataract phenotype in a much later developmental stage (60% at 3 mpf). Similar to mammalian Cx46 and Cx50[90,91], the immunostaining for Cx35.1 (*gjd2b*) appeared around the equatorial region of the lens. We presume that Cx35.1 (*gjd2b*), like other vertebrate lenticular connexins[92,93,100], is essential in zebrafish for lens circulation by facilitating an outward intracellular current of water, ions, and small metabolites. Disruption of this microcirculation system may have led to the accumulation of metabolic waste[91,101,102] inducing the observed scattering and diffraction. As the lenticular immunoreactivity was only modest, further investigation is needed to pinpoint the exact topographic distribution of Cx35.1 (*gjd2b*) throughout the zebrafish lens. Translation of our findings into human cataracts is challenging given that expression of *GJD2* (Cx36) in human lenses has not been reported. This may suggest that *gjd2b* (Cx35.1) has evolved and diverged in function in zebrafish.

Consistent with the negative RE in the *gjd2b* (Cx35.1) mutant fish, in humans, patients with nuclear cataract experience lenticular myopia due to an increased refractive index[103–105]. The

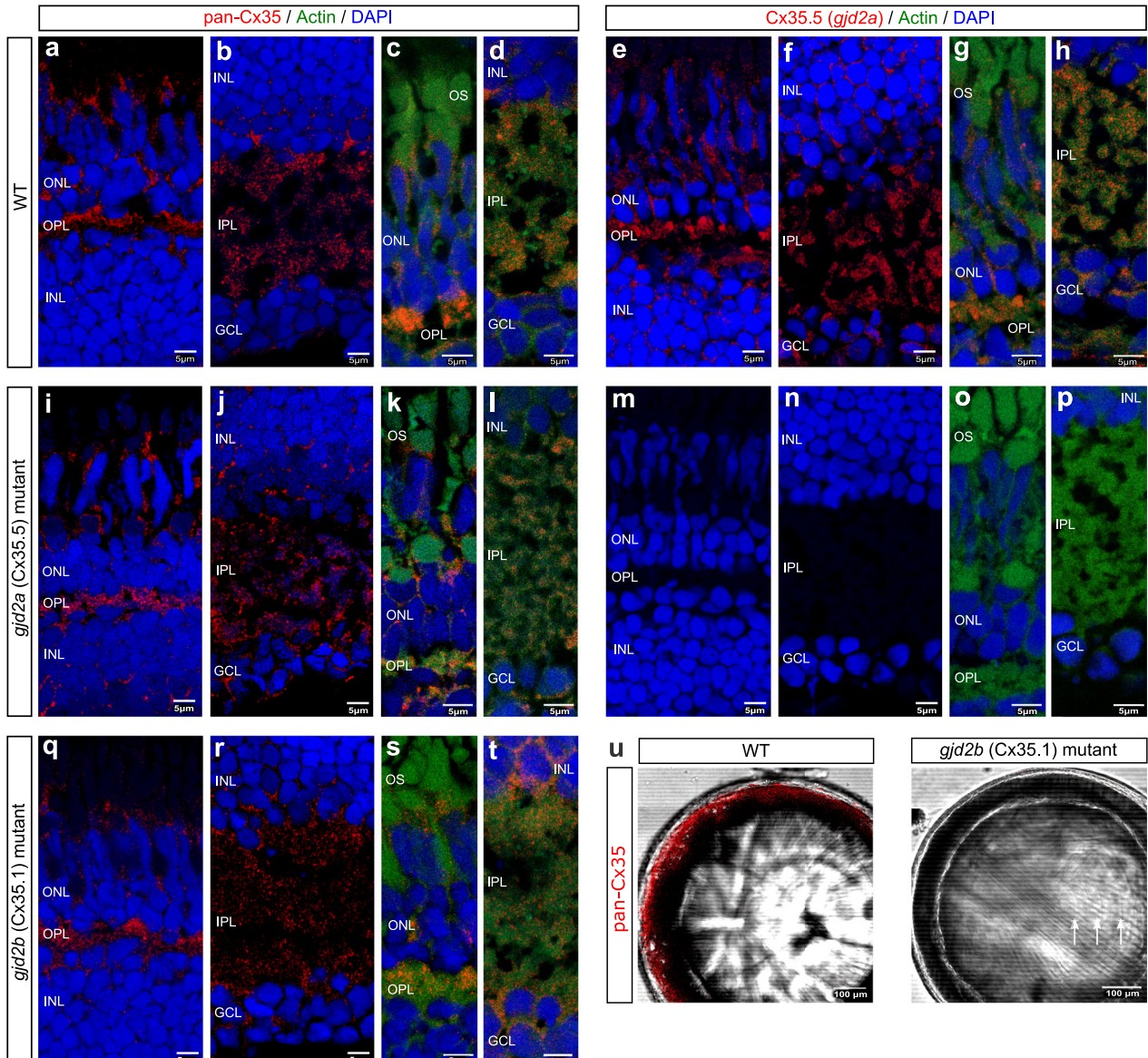

**Fig. 6 Ocular expression of Cx35.5 (gjd2a) and Cx35.1 (gjd2b).** Immunostaining showing the topographic distribution of Cx35.5 (gjd2a) and Cx35.1 (gjd2b) throughout the 2.5 mpf zebrafish retina (**a**)–(**t**) and Cx35.1 (gjd2b) throughout the 6 mpf zebrafish lens (**u**). The retinal sections in the left and right column are stained for respectively anti-pan-Cx35 and anti-Cx35.5 (gjd2a) (both in red) and each row indicates the genotype. DAPI (blue) and anti-actin (green) are used for orientation. The staining in the WT retina for anti-pan-Cx35 (**a**)–(**d**) and the specific anti-Cx35.5 (**e**)–(**h**) reveal generally overlapping patterns of localization in the IPL, OPL, and photoreceptor layer. In the gjd2a (Cx35.5) mutants the anti-pan-Cx35 staining (**i**)–(**l**) supports that this antibody recognizes both Cx35.5 (gjd2a) and Cx35.1 (gjd2b) and reveals the localization of Cx35.1 (gjd2b). The absence of staining in the gjd2a (Cx35.5) mutants with the anti-Cx35.5 antibody (**m**)–(**p**) support the specificity of the antibody and confirms the Cx35.5 (gjd2a) localization presented in (**e**)–(**h**). The presence of staining in the gjd2b (Cx35.1) mutants with the anti-pan-Cx35 antibody (**q**)–(**t**) support that this antibody recognizes both Cx35.5 (gjd2a) and Cx35.1 (gjd2b) and supports the localization of Cx35.5 (gjd2a). **u** Anti-pan-Cx35 immunostaining in isolated 6mpf WT and Cx35.1 (gjd2b) mutant lenses. A modest lenticular appearance at the outer cortical layer can be observed in WT control while absent in gjd2b (Cx35.1) null mutants. The gjd2b (Cx35.1) null mutant lens shows a nuclear ring structure (arrows). Scale bars: 5 μm (**a**)–(**t**), 100 μm (**u**). IPL inner plexiform layer, OPL outer plexiform layer.

severity of the nuclear cataract in those patients correlates strongly with the severity of the myopic shift[103], which is consistent with the progression of both cataract and RE in the gjd2b (Cx35.1) mutants. Given that no biometric changes were observed during the first 3 months, we hypothesize that the initial myopic shift observed with photorefraction may have been induced by the converging properties of the opacifying lens nucleus. During adulthood, the cataract progressed substantially, leading to a degradation of retinal image quality, which may have triggered the late-onset axial elongation observed at 6 mpf. In the

juvenile fish (1.5 and 2 mpf), we did not observe changes in axial length due to the fact that only 25% of the fish at this developmental stage showed a significant cataractous phenotype, reducing the sample size and power at these two particular time points. We further demonstrated light distortion by the cataractous lenses of the gjd2b (Cx35.1) mutant, indicative of a level of image degradation compared to the translucent diffusers used to trigger FDM[60,106–110]. The susceptibility for FDM in adult teleost fish has also been confirmed in tilapia[111]. In conclusion, we postulate that the cataract, negative RE, and late-onset axial

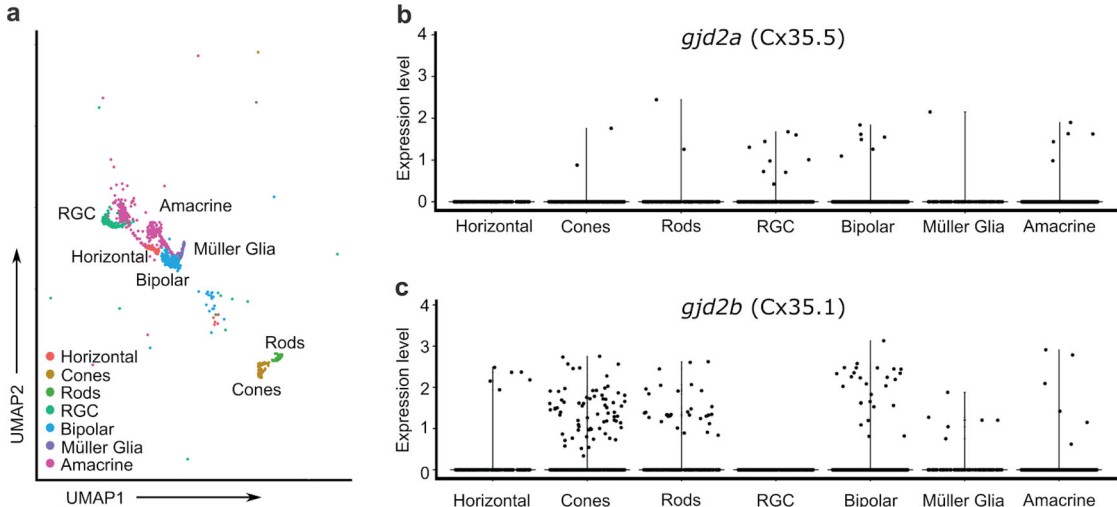

**Fig. 7 Single-cell transcriptome analysis of larval WT zebrafish.** Single-cell transcriptome analysis reveals *gjd2a* (Cx35.5) and *gjd2b* (Cx35.1) expression in a wide variety of retinal cell types. **a** Annotated retinal cell clusters isolated from a single-cell RNA-seq data set of 2–5 dpf larval zebrafish. **b, c** Expression of *gjd2a* (Cx35.5) (**b**) and *gjd2b* (Cx35.1) (**c**) in seven identified retinal cell clusters. Each dot represents an individual cell and its expression level of the indicated gene. UMAP uniform manifold approximation and projection, RGC retinal ganglion cell.

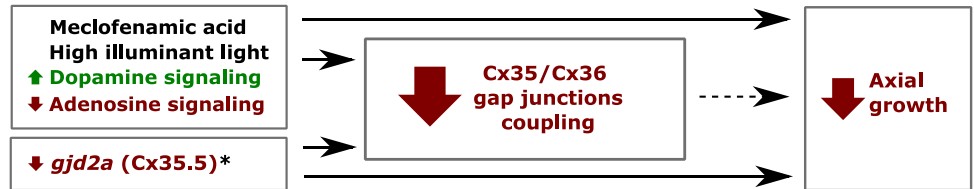

**Fig. 8 Gap junction regulators and the effect on axial growth.** Simplified illustration summarizing the relation between (un)coupling of retinal gap junctions and axial eye growth. We hypothesize, based on reported literature and the outcome of this study (asterisk), that the (un)coupling of retinal gap junctions may play an intermediate role in controlling emmetropization. The exact mechanism of how Cx35/Cx36 uncoupling leads to reduce axial eye growth is unclear (dotted line arrow).

elongation were more an effect of the lenticular and not intra-retinal loss of Cx35.1 (*gjd2b*) gap junctions and that the function of *gjd2b* (Cx35.1) in zebrafish diverges from the function of human *GJD2*. Also, the reduction of the number of eye-tracking movements after the loss of Cx35.1 (*gjd2b*) may have been the result of the emerging cataract rather than an intraretinal signaling defect, as the B-wave amplitude was not altered.

In contrast to Cx35.5 (*gjd2a*), Cx35.1 (*gjd2b*) has previously been reported as a retinal connexin in zebrafish. A large part of the described functional and topographic characteristics of Cx35.1 (*gjd2b*) in databases was based on studies into "Cx35" before discovering Cx35.5 (*gjd2a*)[49,50,70,112]. These studies generally used antibodies such as the cross-reacting anti-pan-Cx35 (MAB3045)[49,50] and anti-Cx36 (Ab298)[50] that binds an epitope conserved between Cx35.5 (*gjd2a*) and Cx35.1 (*gjd2b*). Further, functional studies into light-dependent phosphorylation and coupling of "Cx35" used phosphospecific anti-Cx35 antibodies[70,113] that showed cross-reactivity with homologs of other vertebrates[70]. In this study, we showed specific expression and functionality of *gjd2a* (Cx35.5) in the zebrafish retina. Therefore, we presume that the reported topography, and to some extent, functional characteristics of the former "Cx35" was composed of a shared contribution of Cx35.5 (*gjd2a*) and Cx35.1 (*gjd2b*).

We found that the intraretinal distribution of Cx35.5 (*gjd2a*) and Cx35.1 (*gjd2b*) was similar to each other, and as expected, to mammalian Cx36[19,22,24,25,28,31,57,114–118].

Cx36 containing gap junctions in the photoreceptor layer have previously been described in a cone–cone and rod–cone

configuration[25,31,115–118]. Consistently, we showed the localization of both Cx35.5 (*gjd2a*) and Cx35.1 (*gjd2b*) in the photo-receptor layer and modest expression of *gjd2a* (Cx35.5) and *gjd2b* (Cx35.1) in the rods and cones of the 2–5 dpf larvae. In the OPL of adult zebrafish, we found a dense population of Cx35.5 (*gjd2a*) and Cx35.1 (*gjd2b*) gap junction plaques resembling the distribution in mammalian species with cone rich retinas such as humans and guinea pigs[19,49,70]. The scRNA-seq study showed that *gjd2a* (Cx35.5) and *gjd2b* (Cx35.1), similar to mammalian Cx36 (*GJD2*)[19,24,28,31,57,119], were both expressed in bipolar and amacrine cells. Interestingly, *gjd2b* (Cx35.1) expression was also detected in a number of horizontal cells, similar to Cx35 in carp[120], whereas expression of mammalian Cx36 (*GJD2*) has not been reported in horizontal cells[20,24]. In our scRNA-seq, we detected *gjd2a* (Cx35.5), but not *gjd2b* (Cx35.1), expression in larval ganglion cells. As the somatas in the GCL showed a positive signal for the anti-pan-Cx35 in the *gjd2a* (Cx35.5) mutant, the absence of *gjd2b* (Cx35.1) in ganglion cells in the scRNA-seq study may have been a consequence of the early stage in development or the limited total retinal cell count in the data set. Similarly, Cx36 (*GJD2*) has been found in dendrodendritic gap junctions between alpha-type ganglion cells[22,28,114] and ganglion-to-amacrine cells[119]. It is worth noting that given the limited total retinal cell count of the scRNA-seq data, caution is required when making broad interpretations of the data.

In humans, two common genetic variants downstream of *GJD2* (Cx36) have consistently been associated with RE (rs634990 and rs524952)[5–15]. In the reported studies, the risk allele has been associated with a myopic RE. However, as observed in other

complex traits, the examination of the coding region of *GJD2* (Cx36) has not revealed coding variants explaining the GWAS signal, and it has been suggested that the identified variants might play a role through gene regulation[4]. Here, we found that the absence of *gjd2a* (Cx35.5) expression in zebrafish leads to a hyperopic phenotype, and we hypothesized that the observed phenotype is the result of the uncoupling of retinal cells. Therefore, one could speculate that in humans, the identified risk variants might up-regulate the expression of *GJD2*, facilitating the coupling of Cx36 (*GJD2*) gap junctions in the eye. This needs further exploration in studies of human ocular tissues, particularly those exploring regulatory elements such as expression (eQTL) or methylation (meQTL) quantitative trait loci[121].

In summary, we have provided an in-depth ocular characterization of zebrafish *gjd2a* (Cx35.5) and *gjd2b* (Cx35.1). Given the phenotypic changes and the gene expression pattern observed in both the *gjd2a* (Cx35.5) and *gjd2b* (Cx35.1) mutants, we conclude that the hyperopic phenotype observed in the *gjd2a* (Cx35.5) mutant is related to the retinal role of Cx35.5 (*gjd2a*). In contrast, the phenotype observed in the *gjd2b* (Cx35.1) mutant is caused by a cataract, suggesting that the function of Cx35.1 (*gjd2b*) in zebrafish diverges from its human counterpart. Our results suggest that the cellular function of Cx35.5 (*gjd2a*) is most similar to mammalian Cx36 (*GJD2*) in terms of retinal localization, functionality (based on the alterations in B-wave potential), and the potential regulation of the axial length. However, the exact molecular mechanism underlying the reduced axial length after *gjd2a* (Cx35.5) depletion is yet to be determined. Studies examining, for example, the retinal cell-specific transcriptome of the *gjd2a* (Cx35.5) mutant may help understand the retina-to-sclera signaling pathway that follows the lower degree of cellular coupling, and as a result, inhibition of axial growth. Our study demonstrates that zebrafish is a suitable model to study REs. Furthermore, the *gjd2a* (Cx35) mutant provides unique opportunities for exploring the relationship between gap junction uncoupling and eye growth and promises to generate insights into the biological mechanism underlying human REs.

## Methods

**Fish lines and housing.** The *gjd2a* (*gjd2a^{fh437}*) and *gjd2b* (*gjd2b^{fh454}*) mutant lines were provided by Dr. Adam Miller from the Institute of Neuroscience at the University of Oregon. The lines were generated using TALENs by targeting the first exon[32]. Furthermore, to minimize potential differences due to genetic background, we used (for both *gjd2a* (Cx35.5) and *gjd2b* (Cx35.1)) the offspring of a heterozygous incross (e.g., *gjd2a^{+/−}* x *gjd2a^{+/−}*), to generate the WT control and mutant lines studied here. The *lrp2^{−/−}* (*lrp2^{mw1}*) mutant line[42,43] was kindly provided by Prof. Brian A. Link (Cell Biology, Neurobiology & Anatomy, Medical College of Wisconsin). Stable zebrafish lines were genetically validated by Sanger sequencing, and raised in tanks with a matched population size and constant feeding pattern to minimize the environmental influence of body size in ocular development. During OCT, photorefraction, OKR, and ERG, zebrafish were anesthetized using a 0.016% tricaine methanesulfonate solution (MS222, Sigma Aldrich), buffered to pH = 7.

All animals were raised and treated in accordance with the Dutch animal welfare legislation and the guidelines from the experimental animal health care center (EDC: Experimenteel Dier Centrum) of the Erasmus Medical Center Rotterdam, The Netherlands. All experiments were conducted in accordance with the European Commission Council Directive 2010/63/EU (CCD approval, license AVD 1010020186907). All fish were kept on a 14-h light: 10-h dark cycle at a constant temperature of 28.5 °C.

**SD-OCT.** To rule out variability in eye metrics as a result of alterations in total body length, body sizes were measured at each time point. To isolate the intraocular biometrical changes from extraocular and globally altered body dimensions, only WT control fish within a 10% range from the average mutant body size were included as a control (Supplementary Fig. S1). Furthermore, to minimize potential differences due to genetic background, we used (for both *gjd2a* (Cx35.5) and *gjd2b* (Cx35.1)) the offspring of a heterozygous incross (e.g., *gjd2a^{+/−}* x *gjd2a^{+/−}*), to generate the WT control and mutant lines studied here. 3D compositions of the eye were generated by a Thorlabs SD-OCT 900 nm Ganymade system. The fish were positioned on a rotatable platform and kept under constant anesthesia. The measured eye was positioned unilaterally above the tricaine solution. The total *XYZ*

field of view was $1.7 \times 1.7 \times 2.2$ mm with a pixel depth of 2 μm in the Z-direction. Custom MATLAB software was used to separate the *XYZ* planes, borders were marked manually for each ocular component. The dimensions of the ocular components were calculated by the script based on the refractive index of the corresponding tissue type. The corneal refractive index was set to 1.33[122–124]. The refractive index of the zebrafish lens was calculated from the gradient refractive index as reported by multiple independent studies and set to 1.4[122–127]. The refractive index for the anterior chamber and vitreous chamber was set to 1.34 and the retinal refractive index was set to 1.38[124,128,129].

**Eccentric photorefraction.** The refractive state of the fish was measured by a custom eccentric infrared photorefractor[36–38] adjusted for zebrafish (Fig. 2a). The fish were anesthetized using 0.016% tricaine solution. During measurements, the eye was kept underwater to simulate the natural aquatic refractive state in which the corneal contribution is negligible[122–124]. Zebrafish are naturally cycloplegic with fixed pupil size, eliminating the use of mydriasis-inducing medication. The lens position is fixed as a result of a vestigial retractor lentis[43,130–133] and most teleosts have a rigid lens shape[133]. This obviates the need for pharmacological tools to prevent accommodation as used for other species. A USB camera (RICOH, TV LENS 50 mm 1:1,4) was aligned perpendicular to the ocular surface with a slight angle relative to the water surface. This prevented irregularities in the brightness profiles as the result of surface reflections. Custom C++ software was used to measure the slope of the gradient brightness profile in real-time. Hundred independent measurements were averaged for each eye. The slope of the brightness gradient was converted into RE by calibrating the system with ophthalmic lenses (−5 to +10 D range). The resulting plot showed a high coefficient of determination ($r^2 = 0.971$) and a conversion factor of 1.924. The conversion factor was used to convert the slope of the brightness profile into the RE (in Diopter). This RE may have been subject to a small eye retinoscopic artifact[36,39,40] i.e., the infrared light reflects back due to the myelin-rich layer of the retinal nerve fiber layer, while during normal conditions the light is optimally focused on the light-sensitive part of the photoreceptors. Considering the visual acuity of zebrafish[134,135] we assumed that WT fish have a refractive status close to emmetropia. The relative RE was calculated by subtracting mutant REs by the baseline RE of the WT group.

**Cataract visualization and quantification.** The presence and proportion of cataract lenses were quantified in coronal (optical) sections at the center of the anterior-posterior axis of 1.5–3 mpf fish. An additional 6 mpf group was analyzed by OCT for the presence of cataracts and relative changes in axial length. For this 6 mpf time-point, the extent of the opacities in the lens was quantified in ImageJ. Binary images of coronal lens sections were created and by automated thresholding, the ratio of opaque pixels was measured. For the ex vivo quantification of the proportion of cataract lenses in the 6 mpf group, a differential interference contrast microscope (Nikon WideField Ti-Eclipse inverted microscope with Coolsnap CCD camera) was used. Lenses were classified as cataractous when the nuclear ring structure was clearly visible.

**Optical properties of the lens.** To evaluate image distortion by cataract, changes in optical capacities were assessed in isolated lenses by transmission measurements and light path visualization. The customized setup for transmission measurements consisted of a transmission meter (Linshang, LS162), emitting light at three wavelengths; IR light (940 nm), UV light (365 nm), and broad-spectrum visible light (380–760 nm). A light sensor detected the relative loss of intensity as a result of optical changes of the lens.

To visualize the light path a customized setup (Fig. 4c) consisting of a 532 nm laser beam, air slit with an absorptive black coating (Acktar, 75 μm × 3 mm), and the standard fluorescent microscope was used. The lenses were placed in a cuvette with a visualization solution containing 10.7% ouzo (3 ml mixed in 25 ml PBS)[136].

**ERG recordings.** Before the ERG recordings, the fish were dark-adapted for at least 30 min and handled under dim, red light illumination. In total, 2.5 mpf fish were subsequently anesthetized using 0.016% tricaine methanesulfonate solution and were placed in a petri dish filled with agarose gel, with the reference electrode placed into the gel and the right eye facing the light source. The spinal cord was severed to stop the heartbeat. Next, a small incision was made with a 25-gauge syringe needle at the edge of the cornea of the right eye, in which the recording electrode filled with E2 medium was placed[137]. Two 100 ms[137–139] light stimuli with a light intensity of ~6000 lux were given with an 8000 ms interval. The response was amplified 10.000 times with a band-pass of 700–0.1 Hz and recorded with the Signal6.03 software (Cambridge Electronic Design Limited). The recordings were baseline corrected, with the baseline signal determined as the average signal before the stimulus was given, during a 50 ms timespan. The average response to the two light stimuli was plotted and the maximal B wave amplitude was calculated and additionally normalized for body length.

**The optokinetic response.** Visual acuity was measured in 3 mpf fish with a custom recording device (Supplementary Fig. S4c) consisting of a computer, LED light (TCAM, Ring Light, 0–100%/12V/6000–7000 K), infrared-emitting diode (XIASONGXIN LIGHT, 9–12 V/10 W/1050 MA), USB camera (Ricoh, TV lens 50

mm 1:1,4), tachometer (Autoleader, NJK-5002C) and an electrical motor (Makeblock, 37MM/DC12,0 V/50 RPM ± 12%/1:90). An infrared 850 nm long-pass filter (Midopt, LP695-46) was used to ensure selective transmission of the infrared light to the camera. The camera was set to 96 frames per second. The contrast of the drum was kept at 100% and the velocity of the drum at 20 degrees per second (d/s). Custom-developed Python software was used to track eye movements in real-time. Fish were briefly anesthetized and fixated dorsally on a small foam platform inside a transparent cylindrical polystyrene water tank. After recovery from anesthesia, fish were exposed to visual stimuli consisting of a grating pattern of alternating black and white stripes. All eyes were pre-stimulated for 5 s with a standard stimulus. Measurements were performed binocularly and multidirectional.

For the first analysis, the spatial frequency was increased in steps of 0.05 cycles per degree (cpd), starting at 0.15 cpd, until an OKR could no longer be elicited. The highest spatial frequency was repeated to verify the loss of the OKR pattern. The response was considered positive when three or more consecutive optokinetic nystagmus patterns were detected in both the temporal to nasal and nasal to temporal direction. The visual acuity was defined as the maximum spatial frequency in cpd that was reached. As a second analysis, the slope of the slow phase of the OKR pattern was quantified and the eye velocity was calculated in d/s. This was done for three different spatial frequencies; 0.15, 0.20, and 0.25 cpd. Calculated values were quantified for the temporally and nasally directed movements. The OKR graphs were analyzed in Python (version 3.8. 0) and the optokinetic gain was determined as the ratio between the eye velocity and drum velocity, in which a gain of 1 indicates a perfect tracking of the moving stimulus. In addition, the number of eye-tracking movements per 10 s interval was quantified for each spatial frequency.

**Histology**. For immunostaining of retinal sections, the eyes of 2.5 mpf fish were dissected and embedded in OCT medium (Scigen, Fisher scientific). The eyes were frozen in nitrogen/penta-butane, and stored at −20 °C. Cryosections of 8 μm were cut laterally and placed on adhesion slides (SuperFrost Plus, Fisher Scientific). Slides were dried with pressurized air and fixated in 4% PFA for 8 min. The slides were washed in PBS and subsequently blocked for 1 h at room temperature in a 5% sheep serum (Sigma-Aldrich) diluted in PBS solution. A wash step of 3 times 5 min in PBS was performed, and the primary antibody (1:100) was applied. After overnight incubation at 4 °C, slides were washed and secondary antibodies applied. For MAB3045 (Millipore) an anti-mouse-cy5 (Sigma-Aldrich) was used and for anti-*gjd2a*/Cx35.5 (Generated by Miller et al. [32]) an anti-rabbit-cy5 (Jackson ImmunoResearch), both 1:200. As a cytoplasmic marker Phalloidin546 (1:200) (ThermoFisher) was used. After 1 h incubation at RT, slides were washed and mounted with ProLong Gold Antifade Mountant with DAPI (ThermoFisher). Imaging was done using a Confocal (Zeiss LSM700) microscope.

For whole-mount imaging of the lens, 6 mpf zebrafish lenses were dissected in PBS and fixed in 1.5% PFA in PBS for 24 h at room temperature. Lenses were washed 3× times 10 min in PBS and permeabilized in a PBS-0,1% Tween-20 solution, overnight at 4 °C, and subsequently incubated with primary antibodies diluted 1:100 in PBS-0,1% Tween-20 and, again, incubated overnight at 4 °C. Lenses were washed three times and secondary antibodies were diluted 1:200 in PBS-0,1% Tween-20 and incubated overnight at 4 °C. For each genotype, the immunostaining was performed twice. For imaging, lenses were fixated in 1.8% low melting point agarose (Invitrogen) covered with PBS. A confocal microscope (Leica SP5 AOBS) with a water immersion objective (HCX PL APO 20× NA 1.0) and 561 nm laser DIC were used to record z-stacks of the lenses. Subsequently, coronal optical slices were evaluated over the anterior-posterior axis.

**Single-cell RNA-sequencing**. Details as described in Farnsworth et al.[52] and modified slightly as noted below. Briefly, cells were dissociated from the Tg(olig2: GFP)vu12 and Tg(elavl3:GCaMP6s) larvae sampled at 1, 2, and 5 days post-fertilization. Dissociated cells were then run on a 10X Chromium platform using 10× v.2 chemistry[140]. To ensure that the full transcript of the Connexin-encoding genes was represented in the dataset, we updated the Ensembl release 97 General Transfer Format (GTF) gene model file by using pooled, short-read RNA-seq data. Using this updated GTF file, we aligned reads to the zebrafish GRCz11_93 genome using the 10X Cellranger pipeline (version 3.1)[140]. We used the Seurat[141,142] V3 software package for R, v3.6.2 using standard quality control, normalization, and analysis steps to cluster and we used 140 principle components to analyze the final 68,766 cells. UMAP analysis was performed on the whole dataset with a resolution of 13.0, which produced 226 clusters and one singleton. Retinal analysis was performed on a selection of seven clusters based on canonical markers of cell types, capturing a total of 2218 cells. We restricted the retinal analysis to cells originating from the 2 and 5 dpf fish.

**Statistical analysis**. Statistical analyses were performed using R (version 4.0.2). Given that measurements from the two eyes of the fish were available, we used linear mixed models (LMM) to analyze the SD-OCT, photorefraction, DIC, lens intensity, and lens opacity data. The LMM accounts for the correlations between paired eyes of a fish by adding a random intercept. LMM has been described to appropriately account for inter-eye correlation while maximizing power and precision[143].

In all the performed mixed models we controlled for genotype (WT or mutant) as a fixed factor, and for fish (subject) as a random factor. The following outcomes were investigated: axial length, anterior chamber depth, lens diameter, vitreous chamber depth, retinal diameter, and RPE thickness in μm (SD-OCT), RE in Diopters (photorefraction), the number of opaque pixels (lens opacity by OCT) and the percentual loss of transmitted light through the lenses.

Even though LMM also allows the analyses of longitudinal data, individual labeling of fish is challenging. Therefore, we analyzed the studied time points (1.5, 2, 3, 6, and 9 mpf) separately (cross-sectional) and not as longitudinal data.

For the ERG, Optical properties of the lens, OKR, and ex vivo light propagation experiments we used Welch's ANOVA (Prism version 8.4.1).

**Reporting summary**. Further information on research design is available in the Nature Research Reporting Summary linked to this article.

## Data availability
All data generated or analyzed during this study are included in this published article and its supplementary information files. Any additional (raw) data are available from the corresponding author on request. Data used in the scRNA-seq study is publicly available at 'https://www.adammillerlab.com/resources-1' and 'http://cells.ucsc.edu/?ds=zebrafish-dev'.

## Code availability
Any custom code and software used in this manuscript can be obtained from the corresponding author on reasonable request.

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

## Acknowledgements
We would like to thank Prof. Brian A. Link (Cell Biology, Neurobiology & Anatomy, Medical College of Wisconsin) for sharing the *lrp2^mw1* mutant fish, Prof. John O'Brien (Ruiz Department Of Ophthalmology & Visual Science, McGovern Medical School) for sharing his expertise on retinal connexins, and Prof. Maarten Kamermans (Netherlands Institute for Neuroscience) for sharing his expertise in retina and zebrafish. Also, we thank Johan Crins, Nina Kokke, Lotte Martens, Ewoud Ruighaver, and Joanna Świer-kowska for their technical support. Funding for this work was supported by the European Research Council (ERC) under the European Union's Horizon 2020 Research and Innovation Program (grant 648268) and the Netherlands Organization for Scientific Research (NWO, grant 91815655 and VidW.1154.18.046) and Prof Dr. Henkes Stichting. This work was supported by the following foundations: Glaucoomfonds and Landelijke Stichting voor Blinden en Slechtzienden (LSBS) that contributed through UitZicht (project 2018–29). The funding organizations had no role in the design or conduct of this research. They provided unrestricted grants. This work was also supported by the NIH National Institute of General Medical Sciences, Genetics Training Grant T32GM007413 to R.M.L., and the NIH Office of the Director R24OD026591 and the NIH National Institute of Neurological Disorders and Stroke R01NS105758 to A.C.M.

## Author contributions
W.H.Q., M.M.S., R.W., C.C.W.K., and A.I.I., contributed to the design of the study. W.H.Q., K.C.D.T., and M.H., performed the experiments and screened the mutants. B.H. J.W. designed the OCT processing and analysis software. F.S. designed the photorefractor software. H.M.d.G. and W.H.Q., conducted DIC microscopy. A.C.M. and R.M.L., provided mutant lines, antibodies and performed single-cell RNA-sequencing studies. E.d.V., S.B. and E.v.W., performed electrophysiological studies. R.W., C.C.W.K. and A.I.I., supervised the study. W.H.Q. and A.I.I. drafted the manuscript. A.C.M., C.C.W.K., E.d.V., F.S., M.M.S., R.M.L, and R.W., critically reviewed the manuscript. All the authors read and agreed on the final draft of the manuscript.

## Competing interests
The authors declare no conflict of interest.
