## [Transparent Peer Review File · Communications Biology]

Reviewers' comments:

Reviewer #1 (Remarks to the Author):

Scope of manuscript:

Previous GWAS experiments established a link between the locus of the connexin gene GJD2 in humans with myopia. Here, the authors investigate the effects of loss-of-function mutations on a class of homologous zebrafish connexin proteins, Cx35.5 and Cx35.1, encoded by the genes *gjd2a* and *gjd2b*, respectively. The authors found the anteroposterior axis to be unchanged in *gjd2b* mutants, but reduced in *gjd2a* mutants when compared with wild-type controls controlled for body size. The authors confirmed effects on on-bipolar cells in *gjd2a* mutants in that there was a reduction in B-wave amplitudes, similar to measurements of *gdj2* nullizygous mice. However, rather than myopia, mutations in *gjd2a* resulted in hyperopia and bifocal in *gjd2b* mutants. Visual impairment was also verified by optokinetic response reductions in mutants compared with wild-type fish. These data are consistent with the hypothesis that elevated Cx35.5 activity leads to myopia.

The *gjd2b* mutants developed lenticular cataracts similar to other connexin mutants, drastically affects light refraction. The mutants were found to have no axial length defects at young ages, but rather increased axial length at older ages.

The authors also validate anti-pan-Cx35 and anti-Cx35.5 antibodies in frozen sections and characterize expression of *gjd2a* and *gjd2b* in single-cell RNA sequencing.

Reviewer impression:

The manuscript is competently conceived research of novel findings, and measurements are well quantified and statistically analyzed. The conclusions are within reason and speculation is clearly stated as such.

Minor issues.

1) Construction of Supplementary tables. Supp Table 1 and 2 each contain a column labeled "beta" which seems to be a difference in measurement calculated mutant length minus the wild-type length. A morphometric difference is often instead called delta, similar to other mathematical differences.

2) In Supplemental Table 2 the 1.5-month measurements for the *gjd2b*^{-/-} do not appear to match those of the Figure 1c or agree with the narrative. Figure 1c appears to show a mean axial length less than 1000 μm , indicating an error in some of the cells of the table, including that which lists the axial length mean as 1225 μm .

3) Were the Wild-type fish used in Fig.1 siblings of those mutant fish to which they were compared or simply sized-matched T/AB strain?

4) Line 250. Please spell out acronym "form-derived myopia" before using FDM.

5) Line 327. Larvae is plural and needs no "s".

Reviewer #2 (Remarks to the Author):

This is a delightfully complex manuscript filled with unexpected results. Motivated by the association of non-coding, intergenic SNPs near the GJD2 locus with refractive error, the authors have investigated the roles of Connexin 35 isoforms Cx35.5 (*gjd2a*) and Cx35.1 (*gjd2b*) in development of

refractive error in zebrafish. Among the surprising outcomes were opposite effects of knockout of *gjd2a* and *gjd2b* on refractive error, development of cataracts in *gjd2b* knockouts, and apparently extensive expression of Cx35.5 in the retina. The majority of results are convincing and the discussion is extensive.

I have very few criticisms of the manuscript. The one topic that I feel was incompletely discussed was the retinal function studies with ERG and OKR. It is striking that *gjd2a* knockout produced such a large effect in the ERG study, while *gjd2b* knockout had a larger effect in OKR spatial frequency response and number of eye tracking movements. And yet only *gjd2a* mutation affected eye growth in the first 3 months?

The authors have mentioned that the relationship of *gjd2* with myopia has become a popular topic. They should consider a recent study by Banerjee et al 2020: <https://pubmed.ncbi.nlm.nih.gov/32547367/>, who found that form deprivation myopia caused an increase in Cx36-mediated coupling. While the experiment is opposite, the mechanism may be similar.

Minor comments:

1. Results, lines 99-100: What is meant by RPE "diameter"? Is this RPE thickness?
2. ERG methods, line 456: Was the stimulus really 100 ms? That seems much too long to produce a classic brief flash ERG response.

Reviewer #3 (Remarks to the Author):

Strong genetic evidence (genome-wide-association-studies) implicates the gap junction protein delta2 gene (GJD2) in human myopia (near-sightedness), although the functional mechanism of action remains unknown. GJD2 encodes a 36-kDa protein called connexin 36 (Cx36), a key component of neuronal gap junctions.

The present study investigated mutants deficient for the GJD2 homologous genes in zebrafish (*gjd2a* and *gjd2b*) and employed sophisticated technical methodology to investigate alterations in ocular biometry, providing direct evidence for the involvement of these gap junction proteins in the development of refractive errors in adult zebrafish. However, the phenotype of the *gjd2a* mutant was a progressive onset of hyperopia due to decreased axial length (opposite of the refractive error of myopia, which is associated with increased axial length), and the *gjd2b* mutants developed lenticular cataracts and myopia.

The zebrafish GJD2 genes are clearly implicated in ocular development in that deficits in the Cx36 homologous proteins result in refractive errors of various types. These results provide a convincing demonstration of the multiple functional roles for connexin 36 proteins in regulating ocular development and growth.

General comments:

The ocular biometric, visual behavior, and retinal electrophysiological methods and data presented on juvenile and adult zebrafish are exceptionally strong. In contrast, the single cell RNA sequencing data from 2 to 5 day-old embryos are problematic. Only a few retinal cells of each neuronal type apparently express *gjd2a* and *gjd2b* genes at this early stage; these results add no insights to the analysis and should be omitted.

Specific comments:

Line 88: It is unclear whether the gjd2a and gjd2b mutations altered overall body growth.

Line 94/Fig.1a: Cannot read the blue text on black background.

Line 101: This measurement is not RPE diameter, but rather thickness (or depth).

Line 102: The statistical tests indicate significant differences between mutant and wild type for these values; if they are measurement errors then the statistical methods must be incorrect.

Line 176: Provide information (in the Methods section) on the sources and specificities of the primary antibodies used.

Line 200/Fig.6q,r,s,t: A more convincing demonstration of the expression of Cx35.5(gjd2a), would be to use the monoclonal Ab specific to Cx35.5, rather than the pan-Cx35 antibody, to stain the Cx35.1(gjd2b) mutant retina.

Response to reviewers:

COMMSBIO-20-3269-T (Quint et al)

We would like to express our gratitude to the reviewers for their time and constructive comments. We have modified the text and the figures based on reviewer's comments. Our specific responses to reviewer's questions are described below.

Remarks to the Author

Reviewer #1

Scope of manuscript:

Previous GWAS experiments established a link between the locus of the connexin gene GJD2 in humans with myopia. Here, the authors investigate the effects of loss-of-function mutations on a class of homologous zebrafish connexin proteins, Cx35.5 and Cx35.1, encoded by the genes *gjd2a* and *gjd2b*, respectively. The authors found the anteroposterior axis to be unchanged in *gjd2b* mutants, but reduced in *gjd2a* mutants when compared with wild-type controls controlled for body size. The authors confirmed effects on on-bipolar cells in *gjd2a* mutants in that there was a reduction in B-wave amplitudes, similar to measurements of *gdj2* nullizygous mice. However, rather than myopia, mutations in *gjd2a* resulted in hyperopia and bifocal in *gjd2b* mutants. Visual impairment was also verified by optokinetic response reductions in mutants compared with wild-type fish. These data are consistent with the hypothesis that elevated Cx35.5 activity leads to myopia.

The *gjd2b* mutants developed lenticular cataracts similar to other connexin mutants, drastically affects light refraction. The mutants were found to have no axial length defects at young ages, but rather increased axial length at older ages.

The authors also validate anti-pan-Cx35 and anti-Cx35.5 antibodies in frozen sections and characterize expression of *gjd2a* and *gjd2b* in single-cell RNA sequencing.

Reviewer impression:

The manuscript is competently conceived research of novel findings, and measurements are well quantified and statistically analyzed. The conclusions are within reason and speculation is clearly stated as such.

Question # 1

Construction of Supplementary tables. Supp Table 1 and 2 each contain a column labeled "beta" which seems to be a difference in measurement calculated mutant length minus the wild-type length. A morphometric difference is often instead called delta, similar to other mathematical differences.

Response: We apologize for the confusion. In Supplementary Table 1 and 2 we refer to the β -coefficient. In this case, the β -coefficient represents the effect of the genotype on the ocular axial length. The β -coefficient is derived from linear mixed model used to analyze the data. In

Supplementary Table 1 and 2 we have changed the term “Beta” for “ β -coefficient”, and have adjusted the legend accordingly.

Question # 2

In Supplemental Table 2 the 1.5-month measurements for the *gjd2b*^{-/-} do not appear to match those of the Figure 1c or agree with the narrative. Figure 1c appears to show a mean axial length less than 1000 μm , indicating an error in some of the cells of the table, including that which lists the axial length mean as 1225 μm .

Response: We thank the Reviewer for this observation. We have revised and updated the values in Supplementary Table 2.

Question # 3

Were the Wild-type fish used in Fig.1 siblings of those mutant fish to which they were compared or simply sized-matched T/AB strain?

Response: The WT T/AB fish used in this study are closely related to the mutant *gjd2* fish. They share the same genetic background but are no direct siblings.

To minimize potential differences due to genetic background, we performed (for both *gjd2a* and *gjd2b* lines) the following crossings: First, we outcrossed the *gjd2*^{-/-} mutant line with WT AB fish. Posteriorly, we in-crossed the heterozygous offspring obtaining both WT and *gjd2*^{-/-} mutants. We used the offspring of the heterozygous incross for further breeding.

We have clarified this in the text.

Methods section, line 395:

*“Furthermore, to minimize potential differences due to genetic background, we used (for both *gjd2a* (Cx35.5) and *gjd2b* (Cx35.1)) the offspring of a heterozygous incross (e.g., *gjd2a*^{+/-} x *gjd2a*^{+/-}), to generate the WT control and mutant lines studied here.”*

Question # 4

Line 250. Please spell out acronym “form-derived myopia” before using FDM.

Response: We have defined FDM in the manuscript (see line 254).

Question # 5

Line 327. Larvae is plural and needs no “s”.

Response: We have changed the text accordingly.

Reviewer #2

This is a delightfully complex manuscript filled with unexpected results. Motivated by the association of non-coding, intergenic SNPs near the GJD2 locus with refractive error, the authors have investigated the roles of Connexin 35 isoforms Cx35.5 (*gjd2a*) and Cx35.1 (*gjd2b*) in development of refractive error in zebrafish. Among the surprising outcomes were opposite effects of knockout of *gjd2a* and *gjd2b* on refractive error, development of cataracts in *gjd2b* knockouts, and apparently extensive expression of Cx35.5 in the retina. The majority of results are convincing and the discussion is extensive.

Question # 1

I have very few criticisms of the manuscript. The one topic that I feel was incompletely discussed was the retinal function studies with ERG and OKR. It is striking that *gjd2a* knockout produced such a large effect in the ERG study, while *gjd2b* knockout had a larger effect in OKR spatial frequency response and number of eye tracking movements. And yet only *gjd2a* mutation affected eye growth in the first 3 months?

Response: In the revised version of the manuscript, we have expanded the discussion of the ERG findings. Based on our ERG results, one could speculate that in zebrafish, *gjd2a* (Cx35.5) may play a more profound role in, for example, ON-type bipolar cell signaling compared with *gjd2b* (Cx35.1). On-bipolar signaling is highly involved in shaping the B-wave response. This finding supports our theory that in zebrafish, *gjd2a* (Cx35.5) plays a more crucial role in the retina than *gjd2b* (Cx35.1).

Regarding the *gjd2b* (Cx35.1) mutant, we hypothesized that the emerging cataract was insufficient to induce the required distortion of the light stimulus (6000lux) to provoke changes in B-wave amplitude. However, it could have affected the spatial acuity (acting as an optical barrier during OKR measurements). We believe that the significant reduction in the number of eye-tracking movements is the result of the retinal blurring and image distortion caused by the cataract.

Given that ON-bipolar cell signaling is also involved in spatial vision¹ the reason why depletion of *gjd2a* (Cx35.5) caused an electrophysiological effect on the ERG while no changes in the spatial acuity were observed remains unclear. The difference could lie in the sensitivity of both methods, particularly the OKR.

We have discussed this in the discussion section

Discussion section, line 239:

"..however, the diminished B-wave potentials have been reported earlier in Gjd2 (Cx36) null mice and have been attributed to defects in the ON-bipolar signaling⁵³⁻⁵⁷. It remains unknown why depletion of gjd2a (Cx35.5) was sufficient to provoke an electrophysiological effect on ERG while not significantly altering the spatial acuity measured by the optokinetic response, as ON-bipolar cell signaling is also highly involved in spatial vision⁵⁸."

¹ Sugita Y, et al. Contributions of retinal direction-selective ganglion cells to optokinetic responses in mice. Eur J Neurosci. 2013 Sep;38(6):2823-31. doi: 10.1111/ejn.12284. Epub 2013 Jun 12. PMID: 23758086.

Discussion section, line 321:

“In conclusion, we postulate that the cataract, negative RE, and late onset axial elongation were more an effect of the lenticular and not intraretinal loss of Cx35.1 (gjd2b) gap junctions, and that the function of gjd2b (Cx35.1) in zebrafish diverges from the function of human GJD2. Also the reduction of the number of eye tracking movements after the loss of Cx35.1 (gjd2b) may have been the result of the emerging cataract rather than an intraretinal signaling defect, as the B-wave amplitude was not altered.”

Question # 2

The authors have mentioned that the relationship of gjd2 with myopia has become a popular topic. They should consider a recent study by Banerjee et al 2020:<https://pubmed.ncbi.nlm.nih.gov/32547367/> , who found that form deprivation myopia caused and increase in Cx36-mediated coupling. While the experiment is opposite, the mechanism may be similar.

Response: We thank the Reviewer for pointing out this article. We agree with the Reviewer that the study of Banerjee et al, although conceptually different, also supports a relation between Cx36 coupling and refractive error. We have included this reference in the discussion.

Discussion section, line 274:

“A limited number of studies have explored the phosphorylation state of Cx36 in animal models of myopia. A recent study describes an FDM murine model in which eyelids were sutured for 40 days. After removing the lid suture, mice showing a negative refractive error, were selected for immunohistochemistry studies in which the phosphorylation state of Cx36 was assessed. The authors report increase phosphorylation and coupling of Cx36 between All amacrine cells in the studied eyes and attribute their finding to a compensation mechanism aiming to improve the signal-to-noise ratio caused by the defocused state during lid suture⁸⁹. Other studies exploring the intricate relation between the phosphorylation state of Cx36 (GJD2) in the retina and refractive error are warranted.”

Question # 3

Results, lines 99-100: What is meant by RPE “diameter”? Is this RPE thickness?

Response: RPE thickness is the right terminology, we have changed the text accordingly.

Question # 4

ERG methods, line 456: Was the stimulus really 100 ms? That seems much too long to produce a classic brief flash ERG response.

Response: The stimulus used was 100 ms. As pointed out by the Reviewer an stimulus of 100 ms might seem long, more if that is compared with the maximum stimulus duration used in humans (5 ms) . However, a light stimulus of 100 ms and longer are commonly used in the zebrafish field. **Table 1** shows a summary of the stimuli length used in various zebrafish studies. Two out of the five selected articles used 100ms. This is in accordance with a published ERG

protocol for zebrafish in which is stated that a 100ms stimulus is sufficient for eliciting a- and b-waves ². We have added the most relevant references to the methods section manuscript.

Table 1. Length of stimulus reported in various studies performing ERG

Author	Year of publication	Specie	Developmental stage	ERG (stimuli duration)	PMID
McCulloch DL., et al	2015	human	NA	5ms	25502644
Sirisi S., et al	2014	zebrafish	1-2 mpf (juvenile fish)	100ms	24824219
Dona M., et al	2018	zebrafish	5-7 dpf (larvae)	100ms	29777677
Hughes A., et al	1998	zebrafish	Adult fish	200ms	9839967
Emran F., et al	2010	zebrafish	5 dpf (larvae)	500ms	20224035
Zang J., et al	2015	zebrafish	5 dpf (larvae)	500ms	26246494

Reviewer #3

Strong genetic evidence (genome-wide-association-studies) implicates the gap junction protein delta2 gene (GJD2) in human myopia (near-sightedness), although the functional mechanism of action remains unknown. GJD2 encodes a 36-kDa protein called connexin 36 (Cx36), a key component of neuronal gap junctions.

The present study investigated mutants deficient for the GJD2 homologous genes in zebrafish (gjd2a and gjd2b) and employed sophisticated technical methodology to investigate alterations in ocular biometry, providing direct evidence for the involvement of these gap junction proteins in the development of refractive errors in adult zebrafish. However, the phenotype of the gjd2a mutant was a progressive onset of hyperopia due to decreased axial length (opposite of the refractive error of myopia, which is associated with increased axial length), and the gjd2b mutants developed lenticular cataracts and myopia.

The zebrafish GJD2 genes are clearly implicated in ocular development in that deficits in the Cx36 homologous proteins result in refractive errors of various types. These results provide a convincing demonstration of the multiple functional roles for connexin 36 proteins in regulating ocular development and growth.

Question # 1

The ocular biometric, visual behavior, and retinal electrophysiological methods and data presented on juvenile and adult zebrafish are exceptionally strong. In contrast, the single cell RNA sequencing data from 2 to 5 day-old embryos are problematic. Only a few retinal cells of each neuronal type apparently express gjd2a and gjd2b genes at this early stage; these results add no insights to the analysis and should be omitted.

² Fleisch VC, Jametti T, Neuhauss SC. Electroretinogram (ERG) Measurements in Larval Zebrafish. CSH Protoc. 2008 Mar 1;2008:pdb.prot4973. doi: 10.1101/pdb.prot4973. PMID: 21356789.

Response: In the literature, a limited number of studies explore the gene expression profile of the zebrafish retina at the single-cell level. Given the complexity/heterogeneity of the retinal tissue and the rather specific expression of *gjd2a* (Cx35.5) and *gjd2b* (Cx35.1), we decided to explore the expression profile of both genes in single-cell RNA-sequencing data from a whole-embryo dataset. As mentioned by the Reviewer, and stated in the results and discussion section of the article, only a small fraction of the cells showed a positive expression. Despite this limitation, the data supports the results of our immunohistological studies and indicates that these genes are expressed at an early developmental stage. We agree with the editor and believe the data is a useful resource. In the revised version of the manuscript we have not omitted the single-cell RNA-sequencing data from the article. However, we understand the remark of the Reviewer, and have therefore extended the discussion of the limitations.

Discussion section, line 355:

*“...the absence of *gjd2b* (Cx35.1) in ganglion cells in the scRNA-seq study may have been a consequence of the early stage in development or the limited total retinal cell count in the data set. Similarly, *Cx36* (*GJD2*) has been found in dendrodendritic gap junctions between alpha-type ganglion cells^{22,28,114} and ganglion-to-amacrine cells¹¹⁸. It is worth noting that given the limited total retinal cell count of the scRNA-seq data, caution is required when making broad interpretations of the data”*

Question # 2

Line 88: It is unclear whether the *gjd2a* and *gjd2b* mutations altered overall body growth.

Response: The observations mentioned in line 88 were not quantified and analyzed with statistical tests. Slight size variation within and between tanks was macroscopically observed in both *gjd2a* (Cx35.5) and *gjd2b* (Cx35.1) mutant lines at 2mpf. Therefore we decided to use matched controls instead of normalizing against body length (as sometimes used in zebrafish studies). The eye:body length ratio could have potentially induced a bias towards ametropia when a mutant would be consistently bigger or smaller.

We do understand the confusion by the statement in line 88. Considering that we did not perform a formal statistical test, we adjusted the text in the results section line 88:

From

*“As we observed differences in the mean body sizes of *gjd2a* (Cx35.5) and *gjd2b* (Cx35.1) mutants...”*

To

*“To prevent biases induced by potential differences in the mean body sizes of *gjd2a* (Cx35.5) and *gjd2b* (Cx35.1) mutants...”*

Question # 3

Line 94/Fig.1a: Cannot read the blue text on black background.

Response: In Figure 1, we have changed the color of the text to orange.

Question # 4

Line 101: This measurement is not RPE diameter, but rather thickness (or depth).

Response: We corrected the term “RPE diameter” for RPE thickness.

Question # 5

Line 102: The statistical tests indicate significant differences between mutant and wild type for these values; if they are measurement errors then the statistical methods must be incorrect.

Response: The measurements of the Cornea, ACD, and RPE were challenging due the relative thin thickness of these layers. However we found some significant differences in these ocular components and agree that these cannot be seen as “measurement errors”. We rephrased the sentence accordingly.

Results section, line 101:

“..., however, given the relatively thin cornea (~20-26 μm), RPE thickness (~24-36 μm), and almost non-existent anterior chamber depth (~4-6 μm) (Fig. 1a), measurements were potentially prone to a higher level of inaccuracy.”

Question # 6

Line 176: Provide information (in the Methods section) on the sources and specificities of the primary antibodies used.

Response: We have included the sources and specificities of the primary antibodies in the methods section.

Question # 7

Line 200/Fig.6q,r,s,t: A more convincing demonstration of the expression of Cx35.5(gjd2a), would be to use the monoclonal Ab specific to Cx35.5, rather than the pan-Cx35 antibody, to stain the Cx35.1(gjd2b) mutant retina.

Response: We appreciate the Reviewer's comment and agree that using the monoclonal Ab specific to Cx35.5 is the most direct demonstration for showing Cx35.5 expression. That was our goal and indeed why we included the use of this antibody in panels e-h and m-p of **Fig. 6**. We previously showed that this antibody specifically recognized Cx35.5, did not bind the other closely related Connexins, and most importantly here, there was no cross reaction with Cx35.1 (*gjd2b*)

(Miller et al, eLife)³. Therefore, we feel that panels e-h clearly demonstrate Cx35.5 is expressed as described in the text/figure. As further support of this notion, we demonstrate that in the *gjd2a* mutant retina there is a loss of Cx35.5 monoclonal antibody staining (panels m-p), further validating the specificity of this antibody. A particularly powerful experiment within the logic of this figure would be to have a Cx35.1 (*gjd2b*) specific antibody - unfortunately, our efforts to generate such an antibody failed, and therefore this experiment is not possible at this time. Never-the-less, we believe the combination of the pan-Cx35 and Cx35.5-specific antibodies, in conjunction with the mutant analysis, provides a clear demonstration for the conclusions we draw on the expression of these proteins.

³ Miller AC, Whitebitch AC, Shah AN, Marsden KC, Granato M, O'Brien J, Moens CB. A genetic basis for molecular asymmetry at vertebrate electrical synapses. *Elife*. 2017 May 22;6:e25364. doi: 10.7554/eLife.25364. PMID: 28530549; PMCID: PMC5462537.

REVIEWERS' COMMENTS:

Reviewer #1 (Remarks to the Author):

Previous GWAS experiments established a link between the locus of the connexin gene GJD2 in humans with myopia. Here, the authors investigate the effects of loss-of-function mutations on a class of homologous zebrafish connexin proteins, Cx35.5 and Cx35.1, encoded by the genes *gjd2a* and *gjd2b*, respectively. The authors found the anteroposterior axis to be unchanged in *gjd2b* mutants, but reduced in *gjd2a* mutants when compared with wild-type controls controlled for body size. The authors confirmed effects on on-bipolar cells in *gjd2a* mutants in that there was a reduction in B-wave amplitudes, similar to measurements of *gdj2* nullizygous mice. However, rather than myopia, mutations in *gjd2a* resulted in hyperopia and bifocal deficiencies in different *gjd2b* mutants. Visual impairment was also verified by optokinetic response reductions in mutants compared with wild-type fish. These data are consistent with the hypothesis that elevated Cx35.5 activity leads to myopia.

The authors report that *gjd2b* mutants developed lenticular cataracts similar to other connexin mutants, which drastically affects light refraction. The mutants were found to have no axial length defects at young ages, but rather increased axial length at older ages.

The authors also characterize anti-pan-Cx35 and validate anti-Cx35.5 antibodies in frozen sections and characterize expression of *gjd2a* and *gjd2b* in single-cell RNA sequencing.

Reviewer impression:

The manuscript is competently conceived research of novel findings, and measurements are well quantified and statistically analyzed. The conclusions are within reason and remaining questions are clearly stated as such, including why opposite phenotypes result for the two mutant gap junction proteins of the study, possibly due to whether their loss-of-function results in cataracts or not. This work further supports the idea that some organ shape control mechanisms are influenced by connexins.

Reviewer #2 (Remarks to the Author):

The authors have made a number of small changes to this well-written and largely convincing manuscript that improve its quality. I do still have some minor comments regarding statements made in the manuscript.

1. The discussion refers to Cx36-containing gap junctions in "homotypic" cone-cone and "heterotypic" rod-cone configurations (line 341). A term used to refer to gap junctions between different cell types is 'heterologous.' 'Homotypic' refers to the same connexin isoform on both sides of the gap junction, while 'heterotypic' refers to different connexin isoforms on the two sides of the gap junction. Except for the Lee et al. paper, the cited references did not examine whether gap junctions were in a homotypic or heterotypic conformation, only that they contained Cx36. The Lee et al. study is likely a misleading result of differences in antigen accessibility or post-translational modification, but could possibly represent a species anomaly in guinea pig. A very recent study in mouse found Cx36 to be essential in both rods and cones for the rod-cone gap junction (Jin et al. 2020; *Sci. Adv.* 6(28):eaba7232).

2. In the same part of the discussion (line 343) the authors state that they saw localization of both Cx35.5 and Cx35.1 specifically in cones. I have not seen a figure, or even a statement in the results, that indicates demonstration of the presence of either of these connexins specifically in cones, only that it is in the photoreceptor layer and OPL. Indeed, the tissue shown is in very poor condition, with holes throughout the retina and the connexin labeling delocalized. It would be extremely difficult to

demonstrate a specific co-localization in those specimens.

3. In the Supplementary table 5, showing the top 16 differentially expressed genes in clusters of retinal cell types, I have to take exception to the identification of one or two clusters. The cluster labeled "Amacrine" appears to have genes characteristic of Muller glial cells (slc13a5a, ca14, hayl6, fgf24, etc.) or perhaps a progenitor cell, if it is indeed a retinal cell type. The cluster labeled "unspecified neuron" probably includes some amacrine cells, as it has typical amacrine cell markers (tfap2c, lhx9, gad2, pax10, slc32a1).

Reviewer #3 (Remarks to the Author):

The revised manuscript has adequately addressed the concerns raised by this reviewer.

Reviewer #2 (Remarks to the Author):

The authors have made a number of small changes to this well-written and largely convincing manuscript that improve its quality. I do still have some minor comments regarding statements made in the manuscript.

Remark 1. The discussion refers to Cx36-containing gap junctions in “homotypic” cone-cone and “heterotypic” rod-cone configurations (line 341). A term used to refer to gap junctions between different cell types is ‘heterologous.’ ‘Homotypic’ refers to the same connexin isoform on both sides of the gap junction, while ‘heterotypic’ refers to different connexin isoforms on the two sides of the gap junction. Except for the Lee et al. paper, the cited references did not examine whether gap junctions were in a homotypic or heterotypic conformation, only that they contained Cx36. The Lee et al. study is likely a misleading result of differences in antigen accessibility or post-translational modification, but could possibly represent a species anomaly in guinea pig. A very recent study in mouse found Cx36 to be essential in both rods and cones for the rod-cone gap junction (Jin et al. 2020; Sci. Adv. 6(28):eaba7232).

Response 1: We agree with the reviewer that Jin et al., 2020 is a valuable source showing the expression of mouse Cx36 in rods and cones. This article describes gap junctions between rods and cones as homotypic. We added this reference to the paper as the expression in both rods and cones helps explain the RNA sequencing findings and *gjd2a* and *gjd2b* expression in both rods and cones. We used the homotypic and heterotypic nomenclature correctly, and decided not to use the term heterologous as suggested by the reviewer. Many other sources, including Jin et al., 2020 use homotypic to describe Cx36/Cx36 gap junctions between rods and cones, i.e., different cell types. The following sources confirm this and show that homotypic gap junctions may be present between both different and the same cell types: Takeuchi and Suzumura 2014; Beyer, Ebihara, and Berthoud 2013; Bai, Yue, and Aoyama 2018.

The term heterologous that the reviewer suggests is not widely used. To prevent any further discussion we removed the homotypic and heterotypic terminology completely (line 328-329) as this does not change the interpretation of our findings, and does not add crucial information to the discussion.

Remark 2. In the same part of the discussion (line 343) the authors state that they saw localization of both Cx35.5 and Cx35.1 specifically in cones. I have not seen a figure, or even a statement in the results, that indicates demonstration of the presence of either of these connexins specifically in cones, only that it is in the photoreceptor layer and OPL. Indeed, the tissue shown is in very poor condition, with holes throughout the retina and the connexin labeling delocalized. It would be extremely difficult to demonstrate a specific co-localization in those specimens.

Response 2. We thank the reviewer for referring us to Jin et al., 2020 in comment 1. The work of Jin et al., 2020 helps to explain the expression of the *gjd2a* and *gjd2b* orthologues in both cones and rods. We feel confident, based on the shape of the cells and layers, to point out the OPL and photoreceptor layer in Figure 6. However, we are not able to distinguish cones based on the histology.

We, therefore, removed the statement of the cones and changed the sentence (line 329-331) to:

“Consistently, we showed the localization of both Cx35.5 (*gjd2a*) and Cx35.1 (*gjd2b*) in the photoreceptor layer and modest expression of *gjd2a* (Cx35.5) and *gjd2b* (Cx35.1) in rods and cones of the 2-5dpf larvae.”.

Remark 3. In the Supplementary table 5, showing the top 16 differentially expressed genes in clusters of retinal cell types, I have to take exception to the identification of one or two clusters. The cluster labeled “Amacrine” appears to have genes characteristic of Muller glial cells (*slc13a5a*, *ca14*, *hayl6*, *fgf24*, etc.) or perhaps a progenitor cell, if it is indeed a retinal cell type. The cluster labeled “unspecified neuron” probably includes some amacrine cells, as it has typical amacrine cell markers (*tfap2c*, *lhx9*, *gad2*, *pax10*, *slc32a1*).

Response 3. We thank the reviewer for noting this discrepancy in the annotation. We agree that the genes noted by the reviewer are indicative of Muller glia and amacrine cells. The associated Figures and Tables have been updated, as well as the result section of the manuscript.

References:

- Bai, Donglin, Benny Yue, and Hiroshi Aoyama. 2018. “Crucial Motifs and Residues in the Extracellular Loops Influence the Formation and Specificity of Connexin Docking.” *Biochimica et Biophysica Acta. Biomembranes* 1860 (1): 9–21.
- Beyer, Eric C., Lisa Ebihara, and Viviana M. Berthoud. 2013. “Connexin Mutants and Cataracts.” *Frontiers in Pharmacology* 4 (April): 43.
- Takeuchi, Hideyuki, and Akio Suzumura. 2014. “Gap Junctions and Hemichannels Composed of Connexins: Potential Therapeutic Targets for Neurodegenerative Diseases.” *Frontiers in Cellular Neuroscience* 8 (September): 189.